# Computationally-efficient Graph Modeling with Refined Graph Random Features

## Abstract

We propose *refined GRFs* (GRFs++), a new class of *Graph Random Features* (GRFs) for efficient and accurate computations involving kernels defined on the nodes of a graph. GRFs++ resolve some of the long-standing limitations of regular GRFs, including difficulty modeling relationships between more distant nodes. They reduce dependence on sampling long graph random walks via a novel *walk-stitching* technique, concatenating several shorter walks without breaking unbiasedness. By applying these techniques, GRFs++ inherit the approximation quality provided by longer walks but with greater efficiency, trading sequential, inefficient sampling of a long walk for parallel computation of short walks and matrix-matrix multiplication. Furthermore, GRFs++ extend the simplistic GRFs walk termination mechanism (Bernoulli schemes with fixed halting probabilities) to a broader class of strategies, applying general distributions on the walks' lengths. This improves the approximation accuracy of graph kernels, without incurring extra computational cost. We provide empirical evaluations to showcase all our claims and complement our results with theoretical analysis.

## 1 Introduction & Related Work

Graph modeling plays an important role in several applications of machine learning (ML), such as anomaly, community and fraud detection (Kim et al., 2022; 2026; Beutel et al., 2015; Noble & Cook, 2003; Li et al., 2025; Dong et al., 2025; Chen et al., 2024; Liu et al., 2023), recommender systems (Yang et al., 2025; 2023; Deng et al., 2022; Gao et al., 2023), and computational biology (Mao et al., 2024; Banerjee & Jost, 2009; Zhang et al., 2024). As for Euclidean data, research on graph modeling has spurred the development of many hard-coded/learnable or parameterized (Yanardag & Vishwanathan, 2015) classes of *kernels* (similarity functions), defining relationships between nodes in the graph (Smola & Kondor, 2003; Kondor & Lafferty, 2002) or between the graphs themselves (Vishwanathan et al., 2010; Shervashidze et al., 2009).

In this paper, we focus on graph node kernels $K : V(G) \times V(G) \to R$ defined on the vertices V of a given graph G, where the similarity between the nodes is measured via their relationship in the graph – e.g how well-connected the nodes are. Common examples of such kernels include the *d-regularized Laplacian*, *diffusion process*, *p-step random walk*, and *inverse cosine* kernels (Smola & Kondor, 2003; Choromanski, 2023). Computing the corresponding Gram matrices $\mathbf{K}(G) = [K(v_i, v_j)]_{i,j=1}^{N}$ for $v_1, ..., v_N \in V(G)$ tends to be expensive, since this often requires operations of time complexity cubic in the number of graph nodes $N$. For this reason, research has been dedicated to developing efficient approximation strategies. One common approach is to rewrite the graph kernel as a product of two lower rank matrices, linearizing the graph kernel values with some mapping $\phi : V \to \mathbb{R}^m$ as follows:

$$\widehat{K}(v_i, v_j) = \phi(v_i)^\top \phi(v_j). \tag{1}$$

This low-rank factorization unlocks efficient computations with the corresponding kernel matrices. In particular, matrix-multiplication operations no longer require explicit materialization of $\mathbf{K}$, exploiting the associativity of matrix multiplication. However, until recently this approach was restricted to ad-hoc learnable graph kernels defined implicitly via learnable $\phi$ (Wu et al., 2019), rather than approximations of the specific classes listed above.

A series of recent papers proposed a new mechanism called *Graph Random Features* (GRFs) (Choromanski, 2023; Reid et al., 2023; 2024b;a; 2025). GRFs provide unbiased approximation of the classes of graph kernels listed above, with probabilistic mappings $\phi$ obtained via graph random walks. For every graph node $v$, GRFs build a scalar field on the subset of RW-reachable graph nodes V(G) via incremental (kernel-dependent) updates of the field in the visited nodes. This field is then mapped to a node-embedding $\phi(v)$, encoding the relationship of the node to the entire graph G. Since $\phi(v)$ is probabilistic, it is referred to as the *graph random feature* corresponding to $v$. Though originally introduced to approximate kernels defined between pairs of graph nodes, GRFs were recently *lifted* for unbiased approximation of kernels defined between pairs of graphs (Choromanski et al., 2025).

In this paper, we propose a new class of GRF for efficient and accurate computations involving graph kernels defined on the nodes of the graph, that we refer to as *refined GRFs*, or GRFs++. GRFs++ resolve some of the long-standing challenges that regular GRFs face, such as: difficulty in modeling relationships between more distant nodes. They also reduce the dependence on the longer random walks in the graphs, the workhorse mechanism of the regular GRFs. This is done via the newly-proposed *walk-stitching* technique, where several shorter walks are concatenated to emulate the mechanism of conducting longer random walks. By applying this techniques, GRFs++ inherit the approximation quality provided by longer walks, yet via a much more efficient method, effectively trading sequential and less computationally-efficient mechanism of conducting a long walk for a parallel computation of short walks and matrix-matrix multiplications. Furthermore, GRFs++ extend simplistic GRFs' walk-termination mechanism leveraging standard Bernoulli schemes with fixed halting probabilities into a class of strategies applying general distributions on the walks' lengths and maintaining unbiasedness of regular GRFs. This leads to more accurate approximation of the graph kernels under consideration, and with no extra computational cost. We provide empirical evaluations, showcasing all the claims, and complement our results with the theoretical analysis.

This paper is organized as follows:

1. In Sec. 2, we present the refined GRFs++ mechanism, introducing the walk-stitching technique (Sec. 2.2.1), a general termination strategy (Sec. 2.2.2), and their connection to higher-order de-convolutions. In Sec. 2 (continued in Sec. 3), we also provide an intrinsic connection between finding a particular instantiation of the GRFs++ algorithm for a given graph kernel and higher-order *(de-)convolutions* of the discrete series encoding its kernel matrix as a Taylor series involving powers of the graph's weight matrices.
2. In Sec. 3, we provide theoretical analysis of GRFs++, including its unbiasedness and concentration results. We show that stitching more walks improves approximation.
3. In Sec. 4, we provide thorough experimental evidence comparing GRFs++ to regular GRFs on approximation quality, speed, and several downstream tasks: normal vector field prediction on meshes, clustering and graph classification.
4. We conclude in Sec. 5 and provide all additional results in the Appendix (Sec. A).

## 2 Refined GRFs (GRFs++)

### 2.1 Preliminaries: regular GRFs

We start by providing an overview of the regular GRF mechanism. We take a weighted undirected graph G(V, E, $\mathbf{W} = [w(i,j)]_{i,j \in V}$) with $N$ nodes/vertices, where (1) V is a set of vertices, (2) E $\subseteq$ V $\times$ V is a set of undirected edges ($(i,j) \in$ E indicates that there is an edge between $i$ and $j$ in G), and (3) $\mathbf{W} \in \mathbb{R}_{\geq 0}^{N \times N}$ is a weighted adjacency matrix (if no edge exists then the corresponding weight is zero).

We consider the following kernel matrix $\mathbf{K}_{\boldsymbol{\alpha}}(\mathbf{W}) \in \mathbb{R}^{N \times N}$, where $\boldsymbol{\alpha} = (\alpha_k)_{k=0}^{\infty}$ and $\alpha_k \in \mathbb{R}$:

$$\mathbf{K}_{\boldsymbol{\alpha}}(\mathbf{W}) = \sum_{k=0}^{\infty} \alpha_k \mathbf{W}^k. \tag{2}$$

For bounded $(\alpha_k)_{k=0}^{\infty}$ and $\|\mathbf{W}\|_{\infty}$ small enough, the above sum converges. The matrix $\mathbf{K}_{\boldsymbol{\alpha}}(\mathbf{W})$ defines a kernel on the nodes of the underlying graph. Interestingly, Eq. 2 covers

all the special cases of graph node kernels we explicitly listed in Sec. 1. It also covers functions that are not positive definite, since $(\alpha_k)_{k=0}^{\infty}$ can be chosen arbitrarily. From now on, we will associate graph kernels with sequences $(\alpha_k)_{k=0}^{\infty}$.

GRFs enable one to rewrite $\mathbf{K}_{\boldsymbol{\alpha}}(\mathbf{W})$ (in expectation) as $\mathbf{K}_{\boldsymbol{\alpha}}(\mathbf{W}) \overset{\mathbb{E}}{=} \mathbf{K}_1 \mathbf{K}_2^{\top}$, for independently sampled $\mathbf{K}_1, \mathbf{K}_2 \in \mathbb{R}^{N \times d}$ and some $d \leq N$. This factorization enables efficient (sub-quadratic) and unbiased approximation of the matrix-vector products $\mathbf{K}_{\boldsymbol{\alpha}}(\mathbf{W})\mathbf{x}$ as $\mathbf{K}_1(\mathbf{K}_2^{\top}\mathbf{x})$, if $\mathbf{K}_1, \mathbf{K}_2$ are sparse or $d = o(N)$. This is often the case in practice. However, if this does not hold, explicitly materializing $\mathbf{K}_1 \mathbf{K}_2^{\top}$ enables one to approximate $\mathbf{K}_{\boldsymbol{\alpha}}(\mathbf{W})$ in quadratic (c.f. cubic) time. Below, we describe the base GRF method for constructing sparse $\mathbf{K}_1, \mathbf{K}_2$ for $d = N$. Extensions giving $d = o(N)$, using the Johnson-Lindenstrauss Transform (Freksen, 2021), can be found in (Choromanski, 2023). Each $\mathbf{K}_j$ for $j \in \{1, 2\}$ is obtained by row-wise stacking of the vectors $\phi_f(i) \in \mathbb{R}^N$ for $i \in V$, where $f$ is the *modulation function* $f : \mathbb{R} \to \mathbb{R}$, specific to the graph kernel being approximated. The procedure to construct random vectors $\phi_f(i)$ is given in Algorithm 1. Intuitively, one samples an ensemble of RWs from each node $i \in V$. Every time a RW visits a node, the scalar value in that node (the so-called *load*) is updated, depending on the modulation function.

---

**Algorithm 1 Regular GRFs:** Construct vectors $\phi_f(i) \in \mathbb{R}^N$ to approximate $\mathbf{K}_{\boldsymbol{\alpha}}(\mathbf{W})$

---

**Input:** weighted adjacency matrix $\mathbf{W} \in \mathbb{R}^{N \times N}$, vector of unweighted node degrees (number of out-neighbours) $\deg \in \mathbb{R}^N$, modulation function $f : (\mathbb{N} \cup \{0\}) \to \mathbb{R}$, termination probability $p_{\text{halt}} \in (0, 1)$, node $i \in \mathcal{N}$, number of random walks to sample $m \in \mathbb{N}$.
**Output:** random feature vector $\phi_f(i) \in \mathbb{R}^N$

1: initialize: $\phi_f(i) \leftarrow \mathbf{0}$
2: **for** $w = 1, ..., m$ **do**
3:     initialise: $\texttt{load} \leftarrow 1$, $\texttt{current\_node} \leftarrow i$, $\texttt{terminated} \leftarrow \text{False}$, $\texttt{walk\_length} \leftarrow 0$
4:     **while** $\texttt{terminated} = \text{False}$ **do**
5:         $\phi_f(i)[\texttt{current\_node}] \leftarrow \phi_f(i)[\texttt{current\_node}] + \texttt{load} \times f(\texttt{walk\_length})$
6:         $\texttt{walk\_length} \leftarrow \texttt{walk\_length} + 1$
7:         $\texttt{new\_node} \leftarrow \text{Unif}[\mathcal{N}(\texttt{current\_node})]$         ▷ assign to one of neighbours
8:         $\texttt{load} \leftarrow \texttt{load} \times \frac{\deg[\texttt{current\_node}]}{1 - p_{\text{halt}}} \times \mathbf{W}[\texttt{current\_node}, \texttt{new\_node}]$     ▷ update load
9:         $\texttt{current\_node} \leftarrow \texttt{new\_node}$
10:       $\texttt{terminated} \leftarrow (t \sim \text{Unif}(0, 1) < p_{\text{halt}})$     ▷ draw RV $t$ to decide on termination
11:     **end while**
12: **end for**
13: normalize: $\phi_f(i) \leftarrow \phi_f(i)/m$

---

After all the walks terminate, the vector $\phi_f(i)$ is obtained by concatenation of all the scalars/loads from the discrete scalar field, followed by a simple renormalization. It remains to describe how the kernel-dependent modulation function $f$ is constructed. For unbiased estimation, $f : \mathbb{N} \to \mathbb{C}$ needs to satisfy $\sum_{p=0}^{k} f(k-p)f(p) = \alpha_k$, for $k = 0, 1, ...$ (see Theorem 2.1 in (Reid et al., 2024b)).

### 2.2 FROM GRFs TO GRFs++

#### 2.2.1 WALK-STITCHING MECHANISM

The inherently sequential procedure of constructing random walks is not supported by modern accelerators. This is one of the key weaknesses of regular GRFs. Shortening the walks by increasing $p_{\text{halt}}$ can in principle mitigate this, at the cost of giving up modeling relationships between more distant nodes in the graph; a graph kernel value between two nodes $i$ and $j$ whose corresponding walks do not intersect is approximated by zero.

In GRFs++, we propose a novel *walk-stitching* technique, where several independently-calculated shorter walks are combined to emulate sampling a longer walk. Mathematically,

we unbiasedly approximate graph kernel matrix $\mathbf{K}_{\boldsymbol{\alpha}}(\mathbf{W})$ as:

$$\mathbf{K}_{\boldsymbol{\alpha}}(\mathbf{W}) \overset{\mathbb{E}}{=} \prod_{i=1}^{l} \mathbf{K}_1^{(i)}(\mathbf{K}_2^{(i)})^{\top}, \tag{3}$$

We refer to $l \in \mathbb{N}_+$ as the walk-stitching *degree*. Each $i$ corresponds to one pair of intersecting walks from the regular GRF mechanism. GRFs++ with degree $l = 1$ are equivalent to regular GRFs. A schematic is given in Fig. 1.

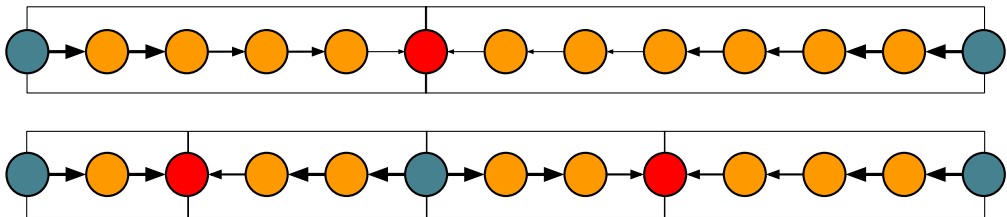

Figure 1: Pictorial description of the *walk-stitching technique*. Each rectangular block corresponds to a random walk and red nodes depict vertices where walks meet. The blue nodes are the communicating ones. The thickness of the arrow, depicting a transition from step $t$ to step $t+1$, indicates the probability that such a transition will occur (a walk can terminate earlier). **Top:** In regular GRFs, two graph vertices communicate via intersecting walks, originating at each vertex. As the nodes become more distant, the probability that such two walks will be constructed decreases. **Bottom:** In GRFs++, two nodes communicate with each other less directly, via proxies (the middle blue node in the picture) and much shorter walks, with lengths that have much higher probability of being realized. The communication is established by stitching several small walks.

Each $\mathbf{K}_j^{(i)}$, for $j \in \{1, 2\}$, is computed as described in Algorithm 1, but the modulation function changes. The following is true:

**Lemma 2.1** (Unbiased walk-stitching and higher-level convolutions)**.** *Suppose that, for each independent instantiation of Alg. 1, the modulation function $f$ satisfies:*

$$\alpha_k = \sum_{p_1 + p_2 + \ldots + p_{2l} = k} f(p_1)f(p_2)\ldots f(p_{2l}). \tag{4}$$

*Then the product $\prod_{i=1}^{l} \mathbf{K}_1^{(i)}(\mathbf{K}_2^{(i)})^{\top}$ provides an unbiased estimation of $\mathbf{K}_{\boldsymbol{\alpha}}(\mathbf{W})$.*

We prove Lemma 2.1 (in fact its more general version) in Sec. 3. The condition from Lemma 2.1 is equivalent to saying that coefficients $\alpha_k$ are obtained via $2l$-level discrete convolution $\overbrace{(f \star f)\ldots(f \star f)}^{l}$ of the modulation function $f$ with itself. Equivalently, the function $f$ must be constructed by $2l$-*de-convolving* sequence $\boldsymbol{\alpha} = (\alpha_k)_{k=0}^{\infty}$ that defines graph kernel.

Interestingly, for several classes of graph kernels this de-convolution can be efficiently calculated. For instance, for graph diffusion kernels with kernel-matrices of the form $\mathbf{K}_{\boldsymbol{\alpha}}(\mathbf{W}) = \exp(\lambda\mathbf{W})$, the correct modulation function for GRFs++ with $l$-degree walk-stitching mechanism is given via a simple expression: $f(p) = \frac{\lambda^p}{(2l)^p p!}$. In Sec. 3, we provide a general mechanism for finding $f$ for more arbitrary $\mathbf{K}_{\boldsymbol{\alpha}}$.

### 2.2.2 GOING BEYOND THE BERNOULLI TRIAL SCHEME

Another key building block of GRFs is the *walk termination mechanism*. In regular GRFs, walk lengths are built incrementally, with walkers terminating independently with probability $p_{\text{halt}}$ at each timestep. This gives the simple update rule in line 10 of Algorithm 1. However, sampling walk lengths from the Bernoulli distribution is not necessarily optimal given fixed computational budget (e.g. fixed average walk length). Here, we propose a very general scheme of RW-length sampling, proposing a simple modification to the update step in Algorithm 1 that improves kernel estimation accuracy.

Take any discrete probabilistic distribution on $\mathbb{N}$: $\mathbf{P} = (P(i))_{i=0}^{\infty}$. We will only assume that: (1) sampling $X \sim \mathbf{P}$ and (2) the computation of $\mathbb{P}(X \geq k)$ for any given $k \in \mathbb{N}$ can be conducted efficiently. We modify Algorithm 1 as follows, to obtain Algorithm 2:

1. The $m$ lengths of walks are sampled: $s_1, ..., s_m \overset{\text{iid}}{\sim} \mathbf{P}$ before line 2.
2. Line 5 is updated as follows, for $\tau(k) \overset{\text{def}}{=} \mathbb{P}(X \geq k)$:

$$\phi_f(i)[\text{current\_node}] \leftarrow \phi_f(i)[\text{current\_node}] + \frac{\text{load} \times f(\text{walk\_length})}{\tau(\text{walk\_length})}.$$

3. In line 8, term $1 - p_{\text{halt}}$ is dropped from the update equation.
4. Line 10 is updated as follows: terminated $\leftarrow \mathbb{I}[\text{walk\_length} \geq s_m]$.

Note that Algorithm 1 is a special instantiation of Algorithm 2, with $\mathbf{P}$ corresponding to the number of consecutive successes of a Bernoulli scheme with failure probability $p_{\text{halt}}$. In Section 3, we show that Lemma 2.1 still holds if Algorithm 1 is replaced by Algorithm 2.

### 2.2.3 PUTTING IT ALL TOGETHER

We are ready to present the complete GRFs++ mechanism, which we will refer to as $(l, \mathbf{P})$-GRFs++, where $l \in \mathbb{N}_+$ and $\mathbf{P} \in \mathcal{P}(\mathbb{N})$ are the hyperparameters of the mechanism. We first construct $(\mathbf{K}_1^i, \mathbf{K}_2^i)_{i=1}^l$, as in Lemma 2.1, but with Algorithm 2 replacing Algorithm 1.

**Option I:** In the most direct approach, the refined random feature vectors are given as rows of the following two matrices $\mathbf{X}, \mathbf{Y} \in \mathbb{R}^{N \times N}$, satisfying $\mathbf{K}_{\boldsymbol{\alpha}}(\mathbf{W}) \overset{\mathbb{E}}{=} \mathbf{X}\mathbf{Y}^{\top}$:

$$\mathbf{X} = \prod_{i=1}^{\frac{l}{2}} \mathbf{K}_1^{(i)}(\mathbf{K}_2^{(i)})^{\top}, \mathbf{Y} = \prod_{i=l}^{\frac{l}{2}+1} \mathbf{K}_2^{(i)}(\mathbf{K}_1^{(i)})^{\top}, \text{ if } l \text{ is even} \tag{5}$$

$$\mathbf{X} = \left[\prod_{i=1}^{\lceil\frac{l-1}{2}\rceil} \mathbf{K}_1^{(i)}(\mathbf{K}_2^{(i)})^{\top}\right] \mathbf{K}_1^{(\frac{l+1}{2})}, \mathbf{Y} = \left[\prod_{i=l}^{\lceil\frac{l+3}{2}\rceil} \mathbf{K}_2^{(i)}(\mathbf{K}_1^{(i)})^{\top}\right] \mathbf{K}_2^{(\frac{l+1}{2})}, \text{ if } l \text{ is odd.} \tag{6}$$

We define the product of the empty sequence of matrices as an identity matrix. If all the matrices $\mathbf{K}_j^{(i)}$ are sparse (i.e. contain only linear in $N$ number of nonzero entries; note for instance that for the regular $p_{\text{halt}}$-termination strategy, the average number of those entries is $Nm\frac{1-p_{\text{halt}}}{p_{\text{halt}}}$), then $\mathbf{X}, \mathbf{Y}$ can be computed in time $O(N^2)$ for constant $l$ and are also sparse. This means the refined random feature vectors are sparse, like their regular counterparts.

**Option II:** Like regular GRFs (Choromanski, 2023), the Johnson-Lindenstrauss Transform (JLT)(Freksen, 2021) can be used to reduce the dimensionality of GRFs++, at the cost of sacrificing their sparsity. The formula for matrices $\mathbf{X}, \mathbf{Y}$ is analogous to this from Option I, but with matrices $\mathbf{K}_j^{(i)}$ replaced by their down-projections, obtained with random Gaussian variates. In particular, we take

$$\widehat{\mathbf{K}}_j^{(i)} = \frac{1}{\sqrt{r}}\mathbf{K}_j^{(i)}\mathbf{G}^{(i)}, \tag{7}$$

for independently created Gaussian matrices $\mathbf{G}^{(i)} \in \mathbb{R}^{N \times r}$, with entries taken independently at random from $\mathcal{N}(0, 1)$ and a hyperparameter $r \in \mathbb{N}$. For constant $l, r$, the computation of all $\widehat{\mathbf{K}}_j^{(i)}$ can be done in $O(N^2)$ time (with no sparsity assumption on $\mathbf{K}_j^{(i)}$). Note also, that under this condition, matrices $\mathbf{X}, \mathbf{Y}$ can be computed in time $O(N)$, via matrix associativity property. Since the JLT preserves dot-products in expectation, we conclude that $\mathbb{E}[\widehat{\mathbf{K}}_1^{(i)}\widehat{\mathbf{K}}_2^{(i)}] = \mathbb{E}[\mathbf{K}_1^{(i)}\mathbf{K}_2^{(i)}]$ for each $i$. Thus resulting $\mathbf{X}, \mathbf{Y}$ still satisfy: $\mathbf{K}_{\alpha}(\mathbf{W}) = \mathbb{E}[\mathbf{X}\mathbf{Y}^{\top}]$.

**Option III:** In practice, as for regular GRFs, GRFs++ do not always need to be explicitly constructed. In most applications of random feature methods, one only needs access to products between the (approximate) kernel matrix and vectors, rather than the kernel matrix itself. As such, one only needs to support efficient multiplication algorithm for $\left[\prod_{i=1}^l \mathbf{K}_1^{(i)}\mathbf{K}_2^{(i)}\right] \mathbf{v}$ for any $\mathbf{v} \in \mathbb{R}^N$. This can be done by multiplying with matrices from the

chain $\prod_{i=1}^{l} \mathbf{K}_1^{(i)} \mathbf{K}_2^{(i)}$ or the chain $\prod_{i=1}^{l} \widehat{\mathbf{K}}_1^{(i)} \widehat{\mathbf{K}}_2^{(i)}$ from right to left, exploiting associativity. If we use the setting from Option I, $l$ is constant and individual matrices are sparse, so this can be done in time $O(N)$ (rather than brute-force $O(N^2)$). This is also the case if Option II is applied with constant $r$.

**Parallel computations of random walks in GRFs++:** One of the most attractive computational features of GRFs++ is that one can compute short RWs in parallel for different $i = 1, 2, ..., l$. These are in turn put together by walk-stitching, implicitly constructing longer walks. This gives computational gains compared to regular GRFs, since it avoids explicit, sequential sampling of longer walks.

**Re-using the same set of random walks:** Even though for unbiasedness, different matrices $\mathbf{K}_j^{(i)}$ for $j \in \{1, 2\}$ ought to use independent sets of random walks, we empirically observe that in practice re-using the same set of random walks also works very well. This is especially the case for larger graphs of higher diameter; see Section 4.

**Walk-stitching with general termination strategies:** To see how walk-stitching helps more distant nodes to connect with each other, consider a termination strategy, where the first transition occurs with probability $p_0 = 1$ and consequent transitions occur with probability $p_{\text{next}} < 1$. In such a setting, the probability of regular GRFs emulating any existing walk of length $r \geq 2$, joining two given vertices $i$ and $j$ scales with $p_{\text{next}}$ as $p_{\text{next}}^{r-2}$, whereas for walk-stitching of degree $\lceil \frac{r}{2} \rceil$, this walk will be emulated via GRFs++ with probability lower-bounded by the expression completely independent of $p_{\text{next}}$.

## 3 THEORETICAL ANALYSIS

We are ready to provide a rigorous theoretical analysis of GRFs++. We start by presenting a strengthened version of Lemma 2.1 from Section 2 (proof in App. A.1).

**Lemma 3.1** (Unbiased walk-stitching, higher-order convolutions & general termination). *Lemma 2.1 remains true if Algorithm 1 in its statement is replaced by Algorithm 2.*

Since Algorithm 2 is more general than Algorithm 1 (see Sec. 2.2.2), this also proves Lemma 2.1. The setting with Algorithm 1 and $l = 1$ is equivalent to regular GRFs.

The formula for the mean squared error (MSE) of the GRFs++-based graph kernel estimator with general degree $l \geq 1$ in terms of the individual components $\mathbf{X}_i = \mathbf{K}_1^{(i)} \mathbf{K}_2^{(i)}$ is complicated. However, for degree $l = 2$, it has a particularly compact form, provided below.

**Lemma 3.2** (MSE of the approximation via GRFs++ with $l = 2$). *The MSE of the estimator $\widehat{\mathbf{K}}_{\boldsymbol{\alpha}}(\mathbf{W})$ of the groundtruth graph kernel matrix $\mathbf{K}_{\boldsymbol{\alpha}}(\mathbf{W})$, leveraging GRFs++ with degree $l = 2$ satisfies (proof in the Appendix: Sec. A.2, $\|\|_F$ stands for the Frobenius norm):*

$$\text{MSE}(\widehat{\mathbf{K}}_{\boldsymbol{\alpha}}(\mathbf{W})) \stackrel{\text{def}}{=} \mathbb{E}[\|\mathbf{X}_1 \mathbf{X}_2 - \mathbf{K}_{\boldsymbol{\alpha}}(\mathbf{W})\|_F^2] = \|\mathbb{E}[\mathbf{X}_1^\top \mathbf{X}_1]\|_F^2 - \|\mathbf{K}_{\boldsymbol{\alpha}}(\mathbf{W})\|_F^2 \tag{8}$$

Finally, we show that the approximation of the graph kernel monotonically improves with the GRFs++ degree (for degrees being the powers of two; proof in App. A.3).

**Theorem 3.3.** *If $\widehat{\mathbf{K}}_{\boldsymbol{\alpha}}^{(l)}(\mathbf{W})$ stands for the estimator of the groundtruth graph kernel matrix, leveraging GRFs++ of degree $l$, then the following holds if standard termination strategy is applied:*

$$\text{MSE}(\widehat{\mathbf{K}}_{\boldsymbol{\alpha}}^{(1)}(\mathbf{W})) \geq \text{MSE}(\widehat{\mathbf{K}}_{\boldsymbol{\alpha}}^{(2)}(\mathbf{W})) \geq \text{MSE}(\widehat{\mathbf{K}}_{\boldsymbol{\alpha}}^{(4)}(\mathbf{W})) \geq ... \tag{9}$$

### 3.1 DE-MYSTIFYING $2l$-LEVEL DE-CONVOLUTIONS

Let us assume that the coefficient $\boldsymbol{\alpha} = (\alpha_k)_{k=0}^{\infty}$ defining graph kernel, encode also an analytical function $g : \mathbb{C} \to \mathbb{C}$ of the form: $g(x) = \sum_{i=0}^{\infty} \alpha_k x^k$. Assume furthermore that one can compute $h(x) = g^{\frac{1}{2l}}(x)$ and its Taylor expansion is of the form: $h(x) = \sum_{i=0}^{\infty} \beta_i x^i$. Then it is easy to see that function: $f(p) = \beta_p$ satisfies Equation 4.

The above observation provides a straightforward algorithm for computing modulation function $f$ for GRFs++ with hyperparameter $l$: (1) map the graph kernel under consideration to function $g$, (2) compute its $(2l)^{th}$-root $h$, (3) find Taylor series of $h$ to define $f$.

**Remark 3.4.** *Now we also see why the formula for $f$ in the GRFs++ mechanism corresponding to the diffusion graph kernel is particularly simple for any $l \in \mathbb{N}_+$. The roots of $g : x \to \exp(x)$, that corresponds to that kernel, are trivial to compute.*

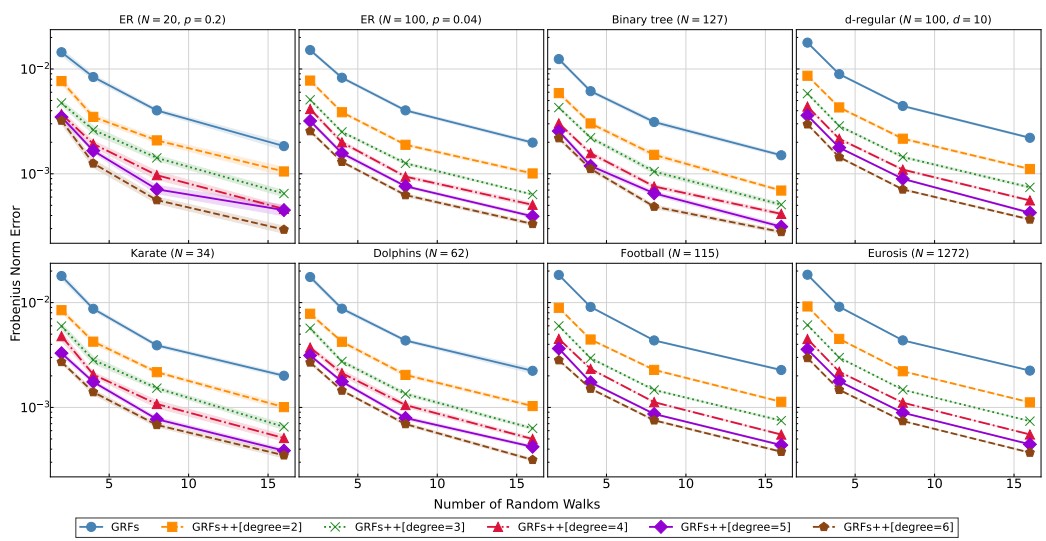

Figure 2: Comparison of different GRF methods for the diffusion kernel estimation. The approximation error (y-axis) improves with the number of walks $m$ (x-axis) and GRF++ provides a sharper estimate than the previous GRF mechanism. The experiment is repeated $s = 10$ times.

## 4 EXPERIMENTS

In this section, we showcase the ability of GRFs++ to efficiently approximate graph node kernels (see Sec. 4.1), including with larger diameters. Furthermore, we show downstream applications of GRFs++ in graph classification, node clustering tasks and normal field predicion on meshes (see Sec 4.2). We use the graph diffusion kernel.

### 4.1 ACCURATE ESTIMATION OF GRAPH KERNELS WITH GRFs++

Following (Reid et al., 2024b), we choose eight graphs of varying sizes: (1) Erdős-Rényi graphs of two sizes, (2) a binary tree, (3) a d-regular graph, and (4) four real world examples (karate, dolphins, football and eurosis). Fig. 2 plots the relative Frobenius norm error of the approximation $\widehat{\mathbf{K}}_{\boldsymbol{\alpha}}(\mathbf{W})$ of the groundtruth kernel matrix $\mathbf{K}_{\boldsymbol{\alpha}}(\mathbf{W})$ with GRFs++ (i.e., $\|\mathbf{K}_{\boldsymbol{\alpha}}(\mathbf{W}) - \widehat{\mathbf{K}}_{\boldsymbol{\alpha}}(\mathbf{W})\|_F / \|\mathbf{K}_{\boldsymbol{\alpha}}(\mathbf{W})\|_F$) against the number of random walks $m$, showcasing improved estimation accuracy with GRFs++. Next, we show that our method can capture long-distance information more accurately than regular GRFs (see Fig. 8 in Appendix). For this task, we select eight diverse graphs from datasets including Peptides (Dwivedi et al., 2022), CIFAR-10 (Dwivedi et al., 2020), Reddit-Binary (Morris et al., 2020), and Geometric Shapes (Yannick-S, 2025). These graphs exhibit varying degrees of sparsity and heterophily, yet all are characterized by large diameters (up to diam = 159). We again compute kernel estimation error (with $p_{\text{halt}} = 0.1$), but now for node pairs that are a specified distance apart. Again, GRFs++ are more accurate. See App. B.1.1 for details.

**New Termination Strategy:** Next, we investigate the benefits a more general (non-Bernoulli) termination strategy, as described in Sec. 2.2.2. Specifically, we employ a halting probability governed by a **Poisson distribution P**. For a fair comparison, the parameters are chosen so that the expected random walk length remains the same as in regular GRFs. Fig. 3 shows that our novel halting strategy improves regular GRF mechanism on a wide range of diverse graphs. We also get a more accurate estimation of kernel values for distant nodes in large diameter graphs (see Fig. 9 in Appendix).

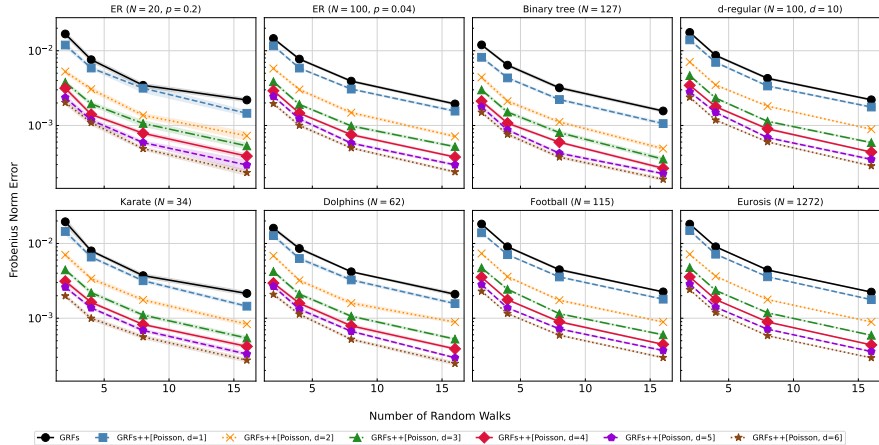

Figure 3: Our novel halting policy based on Poisson distribution provides additional gains over the GRF mechanisms. We run the experiment $s = 10$ times on different graphs of varying sizes.

**Re-using the same set of random walks:** Finally, we conduct an ablation study (see Fig. 4) to pinpoint the benefit of the walk-stitching mechanism itself. In this experiment, rather than sampling new independent walks for each component (as before), we take **the exact same set** of random walks generated for the baseline GRF method and re-use them for each degree of the GRFs++ estimator. This still results in a significant im-

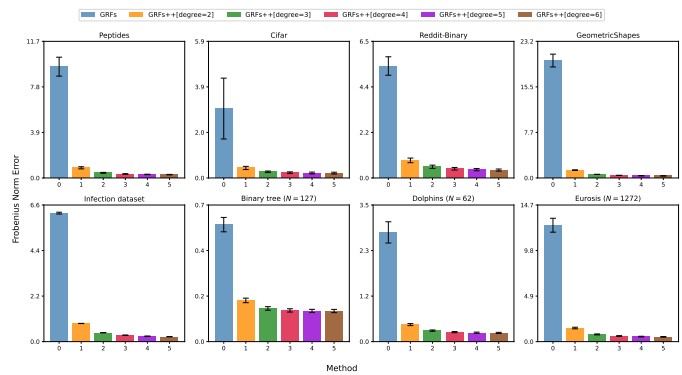

Figure 4: Using the exact same walk as the baseline GRF, repeated multiple times, pinpoints the effectiveness of the walk-stitching algorithm, showing additional computational gains.

provement in the Frobenius norm error for all GRFs++ variants, compared to the regular GRFs baseline (see Tab. 8 for a practical application of a downstream experiment).

**Computational Time:** We generated random graphs with 500 nodes. We evaluated two configurations using base halting probabilities ($p_{\text{halt}}$) of 0.01 and 0.001, which correspond to the regular GRF method (degree $l = 1$). For the GRFs++ methods of degree $l > 1$, we used a scaled

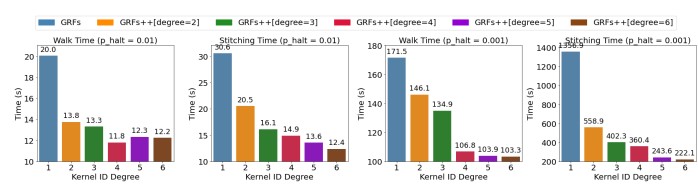

Figure 5: Speed comparison for various GRF-methods: regular GRFs and GRFs++ with different degrees.

halting probability of $p_{\text{halt}} \times l$ to ensure a fair comparison. Fig. 5 presents a computational speed analysis of the GRFs++ algorithm, breaking down its performance into two key components: "Walk Time" (the time required for random walk sampling) and "Stitching Time" (the time for the matrix-matrix operations used in the walk-stitching technique). As the degree increases, both the walk time and the stitching time decreases.

## 4.2 DOWNSTREAM TASKS

In this section, we show the efficacy of our GRFs++ based approximate kernel in various downstream tasks: graph classification, node clustering and vertex normal prediction.

**Graph Classification** : Graph kernels have been widely used for graph classification tasks (Kriege et al., 2020; Nikolentzos et al., 2021). We compare the graph classification results obtained using the approximate kernel from GRF++ with those from the exact diffusion kernel on a wide variety of datasets (Morris et al., 2020). Fig. 6 shows that our method performs on par with the diffusion kernel and outperforms regular GRF (additional details in App. B.2).

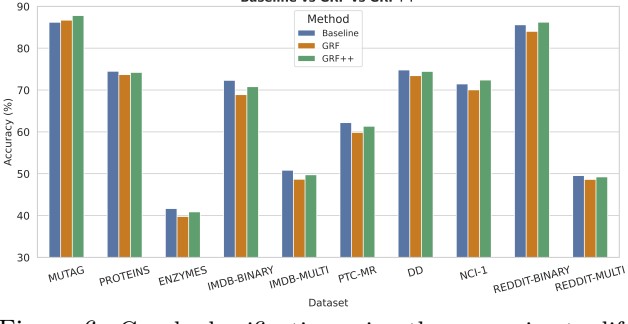

Figure 6: Graph classification using the approximate diffusion kernel from GRF++. Our method performs at par with the baseline diffusion kernel and always beats GRFs.

**Node Clustering**: We also test GRFs++'s utility on the downstream task of node clustering. For this experiment, we perform spectral clustering, implemented using the SciPy library, to group nodes based on the approximated diffusion kernel. We compare the

Table 1: Node Clustering: GRFs++ vs regular GRFs.

| Name | # Nodes | # clusters | GRF | GRF++[d=2] |
|------|---------|-----------|-----|-----------|
| Karate | 34 | 2 | 0.2995 | **0.2585** |
| Dolphins | 62 | 2 | 0.0635 | **0.0323** |
| Polbooks | 105 | 3 | 0.1060 | **0.1033** |
| Football | 115 | 12 | 0.0731 | **0.0362** |
| Databases | 1006 | 6 | 0.3528 | **0.3001** |
| Eurosis | 1272 | 13 | 0.2248 | **0.1304** |

clustering error $E = (\text{\# of wrong pairs})/(N*(N-1))$ of the baseline GRF with our GRF++[degree=2] method. The results, presented in Tab. 9 , show that GRF++ achieves a lower error rate **across all tested datasets** (additional details in App. B.3).

**Normal Prediction** : We test GRF++ on normal vector prediction (mesh interpolation) Every vertex of the graph mesh G with a vertex-set V, is associated with spatial coordinates $x_i \in \mathbb{R}^3$ and a unit normal vector $F_i \in \mathbb{R}^3$. Following (Choromanski et al., 2024), we randomly sample a subset $V' \subset V$ from each mesh with $|V'| = 0.8|V|$ and mask out their vertex normals. Our goal is to predict the vertex normals of each masked vertex $i \in V'$ via: $F_i = \sum_{j \in V \setminus V'} \mathrm{K}(i,j)F_j$, where K is the diffusion kernel. We report the cosine similarity between predicted and groundtruth vertex normals, averaged over all the nodes. We validate GRF++ over 40 meshes of 3D printed objects of varying sizes from the Thingi10K dataset (Zhou & Jacobson, 2016). Additional details are provided in App. B.4. To save space, we show all **40+** mesh results in Tab. 8 in the Appendix. Here we present a few larger meshes in Table 2. GRFs++ provide consistent gains, compared to regular GRFs.

Table 2: Cosine Similarity results for Meshes. GRF++ matches the baseline kernel (BF) and outperforms GRFs. GRF++r, reusing the same random walk, also outperforms GRF.

| MESH SIZE | 5985 | 6577 | 6911 | 7386 | 7953 | 8011 | 8261 | 8449 | 8800 | 9603 |
|-----------|------|------|------|------|------|------|------|------|------|------|
| BF | 0.9194 | 0.9622 | 0.9769 | 0.9437 | 0.9460 | 0.9382 | 0.9196 | 0.9276 | 0.9836 | 0.9766 |
| GRF | 0.9091 | 0.9525 | 0.9682 | 0.9308 | 0.9383 | 0.9233 | 0.9050 | 0.9139 | 0.9778 | 0.9708 |
| GRF++ | 0.9154 | 0.9599 | 0.9751 | 0.9374 | 0.9429 | 0.9321 | 0.9145 | 0.9205 | 0.9820 | 0.9748 |
| GRF++r | 0.9129 | 0.9561 | 0.9701 | 0.9348 | 0.9410 | 0.9269 | 0.9095 | 0.9160 | 0.9805 | 0.9734 |
| Diff | 0.0063 | 0.0074 | 0.0069 | 0.0066 | 0.0046 | 0.0088 | 0.0095 | 0.0066 | 0.0042 | 0.0040 |

## 5 CONCLUSION

We introduced *refined GRFs* (GRFs++) for improved approximation of graph node kernels. GRFs++ address regular GRFs' shortcomings, modeling the relationship between distant pairs of nodes more effectively and introducing novel random walk termination strategies. GRFs++ provide more accurate and more efficient kernel approximation, replacing computationally-inefficient and inherently sequential sampling of long random walks with matrix-matrix operations. We complement our algorithm with theoretical analysis, showing that GRFs++ give unbiased approximation. We provide concentration results, as well as detailed empirical evaluation on a wide variety of graph datasets and tasks.

## 6 REPRODUCIBILITY STATEMENT

The paper provides a clear description of the GRFs++ algorithm. In Sec. 2.1, we present detailed description of the regular GRFs algorithm, namely Algorithm 1 box, that GRFs++ build on. This algorithm is also implemented in the github repository mentioned on the first page of (Reid et al., 2024b). In Lemma 2.1, we explain how Algorithm 1 is used in GRFs++ for the general walk-stitching mechanism. Then in Sec. 2.2.2, we provide detailed explanation of the modification of Algorithm 1 that needs to be conducted in order to support arbitrary termination strategies (points 1-3). For all the experiments, we provided the names of all datasets and graphs used (or exact procedures to construct those graphs, e.g. random Erdős-Rényi graph models with explicitly given probabilities $p$ of edge sampling). All the theoretical statements have all the assumptions clearly stated and the corresponding proofs given (see: Section 2.2.1, Section 3 and Appendix: Section A.1, Section A.2 and Section A.3). Finally, we provide a pointer to the anonymous github repository with code in Appendix: Section B.7.

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

# A APPENDIX

## A.1 PROOF OF LEMMA 3.1

*Proof.* By using similar analysis, as in the proof of Theorem 2.1 from (Reid et al., 2024b), we obtain:

$$\mathbb{E}[\mathbf{K}_1^{(i)}\mathbf{K}_2^{(i)}(a,b)] = \sum_v \sum_{p=0}^{\infty} \sum_{t=0}^{\infty} \mathbf{W}^p(a,v)\mathbf{W}^t(v,b)f(p)f(t)\mathbb{P}[X \geq p]\mathbb{P}[Y \geq t]\frac{1}{\tau(p)}\frac{1}{\tau(t)}, \quad (10)$$

where $X, Y \overset{\text{iid}}{\sim} \mathbf{P}$. Thus, form the fact that pairs $(\mathbf{K}_1^{(i)}, \mathbf{K}_2^{(i)})$ are constructed independently for different $i = 1, ..., l$, we obtain:

$$\mathbb{E}\left[\prod_{i=1}^l \mathbf{K}_1^{(i)}\mathbf{K}^{(i)}(a,b)\right] = \sum_{v_1,v_2,...v_{2l-1}} \sum_{p^{(1)}}^{\infty} ... \sum_{p^{(l)}}^{\infty} \sum_{t^{(1)}}^{\infty} ... \sum_{t^{(l)}}^{\infty} \mathbf{W}^{p^{(1)}}(a,v_1)\mathbf{W}^{t^{(1)}}(v_1,v_2)... $$
$$\mathbf{W}^{p^{(l)}}(v_{2l-2},v_{2l-1})\mathbf{W}^{t^{(l)}}(v_{2l-1},b)f(p^{(1)})f(t^{(1)})...f(p^{(l)})f(t^{(l)}) \quad (11)$$

Therefore, we obtain:

$$\mathbb{E}\left[\prod_{i=1}^l \mathbf{K}_1^{(i)}\mathbf{K}^{(i)}\right] = \sum_{i=0}^{\infty} \left[\sum_{p^{(1)}+t^{(1)}+...+p^{(l)}+t^{(l)}=i} f(p^{(1)})f(t^{(1)})...f(p^{(l)})f(t^{(l)})\right] \mathbf{W}^i \quad (12)$$

That completes the proof, because of Equation 4. $\qquad\square$

## A.2 PROOF OF LEMMA 3.2

We now provide the proof of Lemma 3.2, that we re-state here for reader's convenience:

**Lemma A.1** (MSE of the GRFs++ based graph estimator with GRFs++ degree $l = 2$)**.** *The* MSE *of the estimator* $\widehat{\mathbf{K}}_{\boldsymbol{\alpha}}(\mathbf{W})$ *of the groundtruth graph kernel matrix* $\mathbf{K}_{\boldsymbol{\alpha}}(\mathbf{W})$*, leveraging GRFs++ with degree $l = 2$ satisfies (proof in the Appendix):*

$$\text{MSE}(\widehat{\mathbf{K}}_{\boldsymbol{\alpha}}(\mathbf{W})) \overset{\text{def}}{=} \mathbb{E}[\|\mathbf{X}_1\mathbf{X}_2 - \mathbf{K}_{\boldsymbol{\alpha}}(\mathbf{W})\|_F^2] = \|\mathbb{E}[\mathbf{X}_1^\top\mathbf{X}_1]\|_F^2 - \|\mathbf{K}_{\boldsymbol{\alpha}}(\mathbf{W})\|_F^2 \quad (13)$$

*Proof.* We have the following:

$$\mathbb{E}[\|\mathbf{X}_1\mathbf{X}_2 - \mathbf{K}_{\boldsymbol{\alpha}}(\mathbf{W})\|_F^2] = \mathbb{E}[\|\mathbf{X}_1\mathbf{X}_2 - \mathbb{E}[\mathbf{X}_1\mathbf{X}_2]\|_F^2] = \mathbb{E}[\|\mathbf{X}_1\mathbf{X}_2\|_F^2] - \|\mathbb{E}[\mathbf{X}_1\mathbf{X}_2]\|_F^2 =$$
$$\mathbb{E}[\text{tr}((\mathbf{X}_1\mathbf{X}_2)(\mathbf{X}_1\mathbf{X}_2)^\top)] - \|\mathbb{E}[\mathbf{X}_1\mathbf{X}_2]\|_F^2 = \mathbb{E}[\text{tr}(\mathbf{X}_1\mathbf{X}_2\mathbf{X}_2^\top\mathbf{X}_1^\top)] - \|\mathbb{E}[\mathbf{X}_1\mathbf{X}_2]\|_F^2 =$$
$$\mathbb{E}[\text{tr}(\mathbf{X}_1^\top\mathbf{X}_1\mathbf{X}_2\mathbf{X}_2^\top)] - \|\mathbb{E}[\mathbf{X}_1\mathbf{X}_2]\|_F^2 = \text{tr}(\mathbb{E}[\mathbf{X}_1^\top\mathbf{X}_1\mathbf{X}_2\mathbf{X}_2^\top]) - \|\mathbb{E}[\mathbf{X}_1\mathbf{X}_2]\|_F^2 = \quad (14)$$
$$= \text{tr}(\mathbb{E}[\mathbf{X}_1^\top\mathbf{X}_1]\mathbb{E}[\mathbf{X}_2\mathbf{X}_2^\top]) - \|\mathbb{E}[\mathbf{X}_1\mathbf{X}_2]\|_F^2 = \|\mathbb{E}[\mathbf{X}_1^\top\mathbf{X}_1]\|_F^2 - \|\mathbf{K}_{\boldsymbol{\alpha}}(\mathbf{W})\|_F^2$$

In the series of equalities above, we applied several facts:

1. unbiasedness of the estimator: $\mathbb{E}[\mathbf{X}_1\mathbf{X}_2] = \mathbf{K}_{\boldsymbol{\alpha}}(\mathbf{W})$,

2. standard formula for the scalar variance: $\mathbb{E}[(Z - \mathbb{E}[Z])^2] = \mathbb{E}[Z^2] - (\mathbb{E}[Z])^2$, lifted to the matrix space via Frobenius norm,

3. the following formula: $\|\mathbf{Z}\|_F^2 = \text{tr}(\mathbf{Z}\mathbf{Z}^\top)$, where tr denotes *trace* of the input matrix,

4. cyclic property of the trace: $\text{tr}(\mathbf{A}\mathbf{B}\mathbf{C}\mathbf{D}) = \text{tr}(\mathbf{B}\mathbf{C}\mathbf{D}\mathbf{A})$,

5. the symmetry of $\mathbf{X}_1^\top\mathbf{X}_1$ and $\mathbf{X}_2\mathbf{X}_2^\top$,

6. the equality: $\mathbb{E}[\mathbf{X}_1^\top\mathbf{X}_1] = \mathbb{E}[\mathbf{X}_2\mathbf{X}_2^\top]$ that comes from the definition of $\mathbf{X}_1$ and $\mathbf{X}_2$,

7. independence of $\mathbf{X}_1$ and $\mathbf{X}_2$.

$\qquad\square$

### A.3   Proof of Theorem 3.3

We will now provide a proof of Theorem 3.3.

*Proof.* Without loss of generality, we will assume that $m = 1$. Take two vertices: $i$ and $j$ of a fixed graph G. We will consider two GRFs++ estimators of the value $\mathbf{K_\alpha}(\mathbf{W})[i,j]$ of the graph kernel between them: $\widehat{\mathbf{K}}_{\boldsymbol{\alpha}}^{(2^t)}(\mathbf{W})[i,j]$ and $\widehat{\mathbf{K}}_{\boldsymbol{\alpha}}^{(2^{t+1})}(\mathbf{W})[i,j]$, applying GRFs++ mechanism with degree $2^t$ and $2^{t+1}$ respectively (for $t \geq 0$). Note that since both estimators are unbiased, it only suffices to prove the following:

$$\mathbb{E}[(\widehat{\mathbf{K}}_{\boldsymbol{\alpha}}^{(2^{t+1})}(\mathbf{W})[i,j])^2] \leq \mathbb{E}[(\widehat{\mathbf{K}}_{\boldsymbol{\alpha}}^{(2^t)}(\mathbf{W})[i,j])^2] \tag{15}$$

Note that estimator $\widehat{\mathbf{K}}_{\boldsymbol{\alpha}}^{(2^t)}(\mathbf{W})[i,j]$ can be re-written as:

$$\sum_{\substack{\omega \in \Omega(i,j) \\ i=p_0,v_1,p_1,...,v_{2^t},p_{2^t}=j}} X_{f^{(2^t)}}^{(\omega)}(p_0,v_1)...X_{f^{(2^t)}}^{(\omega)}(p_{2^t-1},v_{2^l})X_{f^{(2^t)}}^{(\omega)}(p_1,v_1)...X_{f^{(2^t)}}^{(\omega)}(p_{2^t},v_{2^l}), \tag{16}$$

where:

1. $\Omega(i,j)$ is the set of all the walks between $i$ and $j$

2. $p_0,...,p_{2^t}$ are some vertices (potentially with repetitions) from $\omega$, visited in that order along the walk $\omega$, as going from $i$ to $j$

3. $X_f^{(\omega)}(a,b)$ is a random variable that is equal to

$$f(l(\omega(a,b)))W(a,b)\left(\prod_{v \in \omega(a,b)} \deg(v)\right)(1 - p_{\text{halt}})^{-l(\omega(a,b))}$$

(for $\omega(a,b)$ denoting vertices on $\omega$ from $a$, but not including $b$, $W(a,b)$ denoting the corresponding product of edge weights and $l(\omega(a,b))$ being the number of edges of the part of $\omega$ from $a$ to $b$) if a random walk from $a$ reaches $b$, as a prefix of $\omega$ starting from $a$ and going to $b$ and is zero otherwise.

4. $f^{(l)}$ is a modulation function for the GRFs++ mechanism of degree $l$.

Similarly, one can write $\widehat{\mathbf{K}}_{\boldsymbol{\alpha}}^{(2^{t+1})}(\mathbf{W})[i,j]$ as:

$$\sum_{\substack{\omega \in \Omega(i,j) \\ i=p_0,v_1,p_1,...,v_{2^t},p_{2^t}=j \\ u_1,u_2,...,u_{2^{t+1}-1},u_{2^{t+1}}}} X_{f^{(2^{t+1})}}^{(\omega)}(p_0,v_1)...X_{f^{(2^{t+1})}}^{(\omega)}(p_{2^t-1},v_{2^t})X_{f^{(2^{t+1})}}^{(\omega)}(p_1,v_1)...X_{f^{(2^{t+1})}}^{(\omega)}(p_{2^t},v_{2^t})$$

$$X_{f^{(2^{t+1})}}^{(\omega)}(p_0,u_1)X_{f^{(2^{t+1})}}^{(\omega)}(p_1,u_3)...X_{f^{(2^{t+1})}}^{(\omega)}(p_{2^t-1},u_{2^{t+1}-1})$$

$$X_{f^{(2^{t+1})}}^{(\omega)}(v_1,u_1)X_{f^{(2^{t+1})}}^{(\omega)}(v_2,u_3)...X_{f^{(2^{t+1})}}^{(\omega)}(v_{2^t},u_{2^{t+1}-1})$$

$$X_{f^{(2^{t+1})}}^{(\omega)}(v_1,u_2)X_{f^{(2^{t+1})}}^{(\omega)}(v_2,u_4)...X_{f^{(2^{t+1})}}^{(\omega)}(v_{2^t},u_{2^{t+1}})$$

$$X_{f^{(2^{t+1})}}^{(\omega)}(p_1,u_2)X_{f^{(2^{t+1})}}^{(\omega)}(p_2,u_4)...X_{f^{(2^{t+1})}}^{(\omega)}(p_{2^t},u_{2^{t+1}}) \tag{17}$$

Denote the sub-sum of the above sum, corresponding to the particular choice of: $p_1,...,p_{2^t},v_1,...,v_{2^t}$ as: $\Psi(p_1,...,p_{2^t},v_1,...,v_{2^t})$.

It suffices to prove that for any two sequences $p_1,p_2,...,p_{2^t},v_1,...,v_{2^t}$ and $p'_1,p'_2,...,p'_{2^t},v'_1,...,v'_{2^t}$, the following holds:

$$\mathbb{E}\Big[\Big(X^{(\omega)}_{f^{(2^t)}}(p_0, v_1)...X^{(\omega)}_{f^{(2^t)}}(p_{2^t-1}, v_{2^l})X^{(\omega)}_{f^{(2^t)}}(p_1, v_1)...X^{(\omega)}_{f^{(2^t)}}(p_{2^t}, v_{2^l})\Big)$$

$$\Big(X^{(\omega)}_{f^{(2^t)}}(p'_0, v'_1)...X^{(\omega)}_{f^{(2^t)}}(p'_{2^t-1}, v'_{2^l})X^{(\omega)}_{f^{(2^t)}}(p'_1, v'_1)...X^{(\omega)}_{f^{(2^t)}}(p'_{2^t}, v'_{2^l})\Big)\Big] \geq \tag{18}$$

$$\mathbb{E}\left[\Psi(p_1, ..., p_{2^t}, v_1, ..., v_{2^t})\Psi(p'_1, ..., p'_{2^t}, v'_1, ..., v'_{2^t})\right]$$

This however follows from the convolutional properties of the modulation function $f$ (Lemma 2.1) and the fact that the product of two $X$-variables corresponding to some walk starting at some fixed vertex of a graph G is not identically zero if and only if one of the walks is a prefix of another one. $\square$

## B  ADDITIONAL EXPERIMENTAL DETAILS

In this section, we provide additional details regarding the experimental setup and present additional results. Moreover we show a plot (Fig. 7) highlighting the speed gains by GRF++ over the baseline kernel. At moderate mesh sizes (2K–4K), GRF++ gives $\sim 1.7 \times -2.2\times$ speedup while for larger meshes (6K–10K), GRF++ consistently delivers $\sim 3\times$ faster runtime. Thus GRF++ scales significantly better as mesh complexity increases.

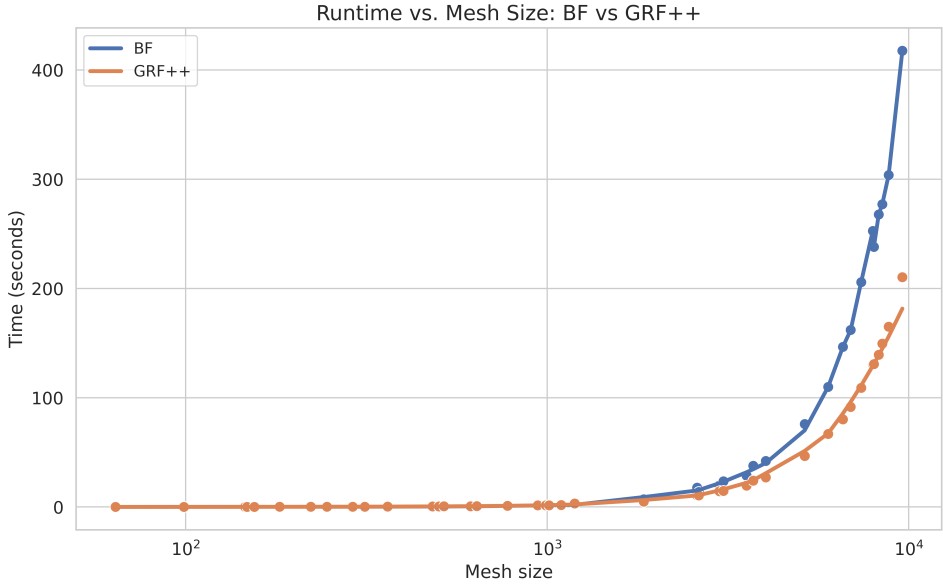

Figure 7: Time comparison between GRF++ and baseline diffusion kernel over various mesh sizes. GRF++ is significantly faster than the baseline as the graphs get larger.

### B.1  ACCURATE ESTIMATION OF GRAPH KERNELS

We follow the exact setup as (Reid et al., 2024b). For computational comparison we used a randomly generated connected graph with 500 nodes. To have fair comparison we derived the relevant $p_{halt} = p_{base} * degree - of - kernel$. We fixed the number of random walks to 256.

### B.1.1  EXPERIMENTS ON GRAPHS WITH LARGE DIAMETERS

In this subsection, we provide details on the graphs used for estimating longer walks. For this task, we pick graphs from Peptides (Dwivedi et al., 2022), CIFAR-10 (Dwivedi et al.,

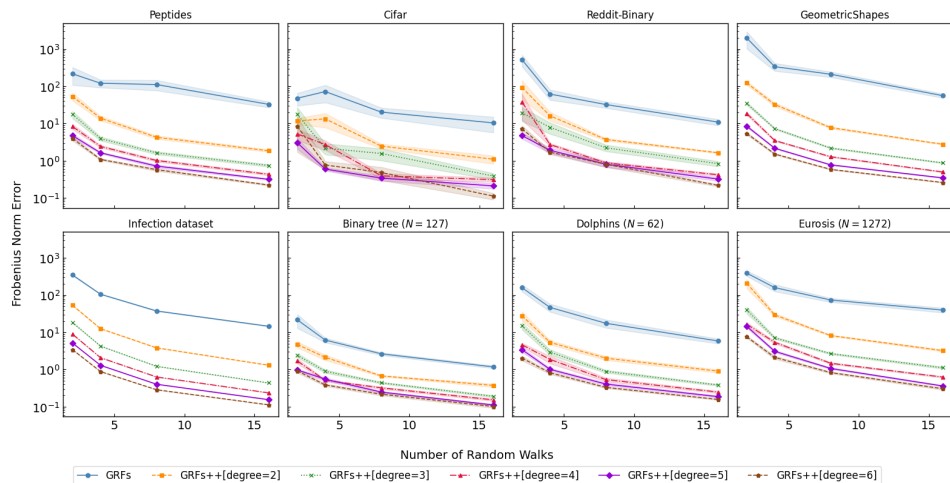

Figure 8: Estimation of the kernel values for distant nodes for the diffusion kernel. GRF++ provides a more accurate estimation in various graphs of large diameters.

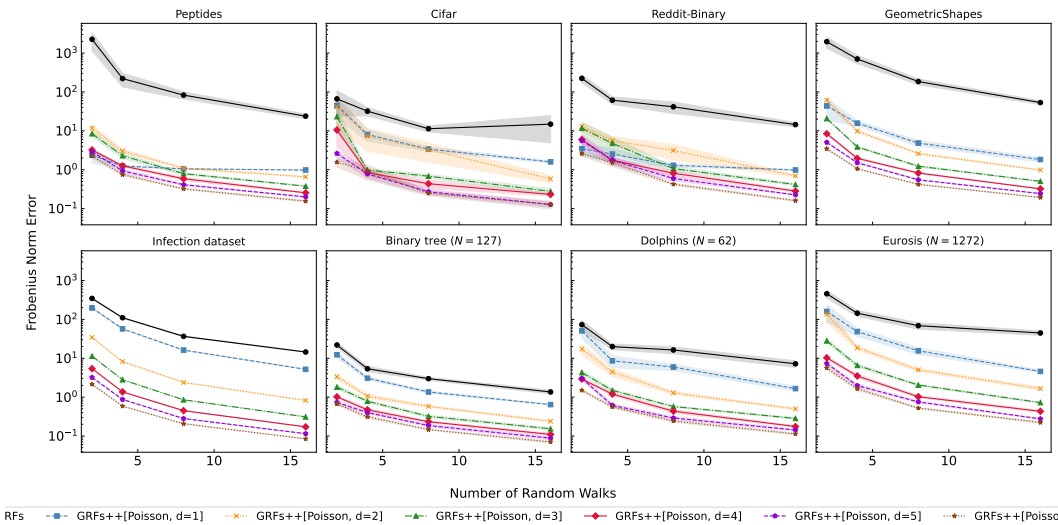

Figure 9: As in Fig. 8, but with Poisson termination strategy activated. This novel halting strategy, proposed in this paper, further improves approximation quality.

2020), Reddit-Binary (Morris et al., 2020), Geometric Shapes (Yannick-S, 2025) as well from the dataset considered by Reid et al. (2024b).

For each of these datasets, we remove isolated nodes and select the graphs with the largest diameters from the subset of the connected graphs. Finally to threshold the graphs to select the longer walks, we compute the shortest path distance via the Floyd-Warshall algorithm. We then select the pair of nodes where the walks are longer than the specified distance away. We then use this information to mask (i.e. zero out) all entries in the diffusion kernel.

We provide the walk threshold for these graphs in Table 3.

## B.2 Graph Classification Experiments

In this subsection we provide additional details about our graph classification experiments. The statistics of our datasets is provided in Table 4 with additional details provided in (Morris et al., 2020). We follow the framework proposed by (Errica et al., 2020) to evaluate the

Table 3: Statistics of datasets used in experiments for estimate the accuracy to capture long walks. The walk length column refers to the fact that we are estimating all walks $\geq k$.

| Dataset | Diameter | # Nodes | Walk Length (k) |
|---|---|---|---|
| Peptides | 159 | 434 | 3 |
| CIFAR | 11 | 128 | 4 |
| Reddit-Binary | 19 | 436 | 5 |
| GeometricShapes | 28 | 864 | 5 |
| Infection | 4 | 500 | 4 |
| Binary Tree | 12 | 127 | 3 |
| Dolphins | 8 | 62 | 4 |
| Eurosis | 10 | 1272 | 4 |

Table 4: Statistics of the graph classification datasets used in this paper.

| DATASETS | # Graphs | # Labels | Avg. # Nodes | Avg. # Edges | # Node Labels | # Node Attributes |
|---|---|---|---|---|---|---|
| MUTAG | 188 | 2 | 17.93 | 19.79 | 7 | - |
| PTC-MR | 344 | 2 | 14.29 | 14.69 | 19 | - |
| ENZYMES | 600 | 6 | 32.63 | 62.14 | 3 | 18 |
| PROTEINS | 1113 | 2 | 39.06 | 72.82 | 3 | 1 |
| D&D | 1178 | 2 | 284.32 | 715.66 | 82 | - |
| IMDB BINARY | 1000 | 2 | 19.77 | 96.53 | - | - |
| IMDB MULTI | 1500 | 3 | 13.0 | 65.94 | - | - |
| NCI1 | 4110 | 2 | 29.87 | 32.30 | 37 | - |
| REDDIT BINARY | 2000 | 2 | 429.63 | 497.75 | - | - |
| REDDIT MULTI-5K | 4999 | 5 | 508.52 | 594.87 | - | - |

performance of the diffusion kernel as well as the approximate kernels obtained by GRF and GRF++. In particular, we use 10-fold cross-validation to obtain an estimate of the generalization performance of the methods.

Finally, we follow the approach by (de Lara & Pineau, 2018) to create graph features by using the smallest $k$-eigenvalues of the corresponding kernels. These features are then passed to a random forest classifier for classification. $k$ is independently for the baseline as well as for GRF and GRF++.

We did a small hyperparameter sweep over $\{.5, .6, .8, .9\}$ to find the width of the diffusion kernel. For GRF and GRF++, we fix the halting probability to be .1 and do a hyperparameter sweep over the number of walks. The degree of GRF++ is chosen to be 2.

Finally we posit our results in the context of various kernel methods as well as GNNs. In particular, we compare our methods against 20 popular kernels like (1) Vertex Histogram kernel (VH), (2) Random Walk kernel (RW), (3) Shortest Path kernel (SP), (4) Graphlet kernel (GR), (5) Weisfeiler-Lehman sub-tree kernel (WL-VH), (6) Weisfeiler-Lehman shortest path kernel (WL-SP), (7) Weisfeiler-Lehman pyramid match kernel (WL-PM), (8) Weisfeiler-Lehman optimal assignment kernel (WLOA), (9) Neighborhood Hash kernel (NH), (10) Neighborhood subgraph pairwise distance kernel (NSPDK), (11) Lovász $\vartheta$ kernel (Lo-$\vartheta$), (12) SVM-$\vartheta$ kernel (SVM-$\vartheta$), (13) Ordered Decompositional DAGs with subtree kernel (ODD-STh), (14) Pyramid Match kernel (PM), (15) GraphHopper kernel (GH), (16) Subgraph Matching kernel (SM), (17) Propagation kernel (PK), (18) Multiscale Laplacian kernel (ML), (19) Core Weisfeiler-Lehman subtree kernel (CORE-WL-VH), and (20) Core Shortest Path kernel (CORE-SP). Note that this kernel computes relationship between graphs whereas our GRF++ computes relationship between nodes in a given graph. Finally we also compare our methods to popular GNNs like (1) DGCNN (Zhang et al., 2018), (2) GraphSAGE (Hamilton et al., 2017), (3) DiffPool (Ying et al., 2018), and (4) GIN (Xu et al., 2019). All these baseline numbers are taken from (Nikolentzos et al., 2021). Our results show that diffusion kernel is competitive across all the datasets. Our GRF++ mechanism maintains similar performance to that of the baseline diffusion kernel while being more computationally efficient.

| | Methods | DATASETS | | | | | |
| --- | --- | --- | --- | --- | --- | --- | --- |
| | | MUTAG | ENZYMES | NCI1 | PTC-MR | D&D | PROTEINS |
| Kernels | VH | 69.1 (± 4.1) | 20.0 (± 4.8) | 55.7 (± 2.0) | 57.1 (± 9.6) | 74.8 (± 3.7) | 71.1 (± 4.4) |
| | RW | 81.4 (± 8.9) | 16.7 (± 1.8) | TIMEOUT | 54.4 (± 9.8) | OUT-OF-MEM | 69.5 (± 5.1) |
| | SP | 82.4 (± 5.5) | 37.3 (± 8.7) | 72.5 (± 2.0) | 60.2 (± 9.4) | 77.9 (± 4.5) | 74.9 (± 3.6) |
| | WL-VH | 86.7 (± 7.3) | 50.7 (± 7.3) | 85.2 (± 2.2) | 64.9 (± 6.4) | 78.7 (± 2.3) | 76.2 (± 3.5) |
| | WL-SP | 81.4 (± 8.7) | 27.3 (± 7.4) | 60.8 (± 2.4) | 54.5 (± 9.8) | 76.0 (± 3.5) | 72.1 (± 3.1) |
| | WL-PM | 88.3 (± 7.1) | 57.5 (± 6.8) | 85.6 (± 1.7) | 65.1 (± 7.5) | OUT-OF-MEM | 75.9 (± 3.8) |
| | WL-OA | 87.2 (± 5.4) | 58.0 (± 5.0) | 86.3 (± 1.6) | 65.7 (± 9.6) | 77.6 (± 3.0) | 76.2 (± 3.9) |
| | NH | 88.3 (± 6.3) | 54.5 (± 3.6) | 84.7 (± 1.9) | 63.4 (± 9.2) | 74.6 (± 3.5) | 75.0 (± 4.2) |
| | NSPDK | 85.6 (± 8.9) | 42.2 (± 8.0) | 74.3 (± 2.1) | 59.1 (± 7.3) | 78.9 (± 4.7) | 72.5 (± 2.9) |
| | ODD-STh | 80.4 (± 8.8) | 32.3 (± 4.8) | 75.2 (± 2.0) | 59.4 (± 9.8) | 76.4 (± 4.5) | 70.9 (± 4.1) |
| | PM | 85.1 (± 5.8) | 43.2 (± 5.3) | 73.5 (± 1.9) | 60.2 (± 8.2) | 77.9 (± 3.7) | 70.9 (± 4.4) |
| | GH | 82.5 (± 5.8) | 37.2 (± 6.6) | 71.0 (± 2.3) | 60.2 (± 9.4) | TIMEOUT | 74.8 (± 2.4) |
| | SM | 85.7 (± 5.8) | 35.7 (± 5.5) | TIMEOUT | 60.2 (± 6.8) | OUT-OF-MEM | OUT-OF-MEM |
| | PK | 76.6 (± 5.2) | 44.0 (± 6.3) | 82.1 (± 2.1) | 65.1 (± 5.6) | 77.7 (± 4.2) | 73.1 (± 4.7) |
| | ML | 87.2 (± 7.5) | 48.5 (± 7.8) | 79.7 (± 1.8) | 64.5 (± 5.8) | 78.6 (± 4.0) | 74.2 (± 4.4) |
| | CORE-WL-VH | 85.6 (± 6.5) | 51.7 (± 7.0) | 85.2 (± 2.2) | 65.5 (± 5.6) | 79.5 (± 3.2) | 76.5 (± 4.4) |
| | CORE-SP | 85.1 (± 6.8) | 39.5 (± 9.3) | 73.8 (± 1.4) | 57.3 (± 9.7) | 79.3 (± 3.8) | 76.5 (± 3.9) |
| | Diffusion | 86.2 (± 4.2) | 41.7 (± 3.6) | 71.5 (± 1.3) | 62.2 (± 5.4) | 74.8 (± 2.2) | 74.5 (± 2.4) |
| | GRF | 86.7 (± 4.4) | 39.8 (± 3.2) | 70.0 (± 1.6) | 59.7 (± 5.8) | 73.4 (± 2.4) | 73.7 (± 2.3) |
| | GRF++ (ours) | 87.8 (± 5.2) | 40.9 (± 3.9) | 72.4 (± 1.2) | 61.3 (± 5.7) | 74.5 (± 2.6) | 74.3 (± 2.8) |
| GNNs | DGCNN | 84.0 (± 7.1) | 46.3 (± 6.3) | 76.4 (± 1.7) | 59.5 (± 6.9) | 76.6 (± 4.3) | 73.2 (± 3.2) |
| | GraphSAGE | 83.6 (± 9.6) | 46.1 (± 5.4) | 76.0 (± 1.8) | 61.7 (± 4.9) | 72.9 (± 2.0) | 74.3 (± 3.8) |
| | DiffPool | 79.8 (± 6.7) | 50.7 (± 8.7) | 76.9 (± 1.9) | 61.1 (± 5.6) | 75.0 (± 3.5) | 72.5 (± 3.5) |
| | GIN | 84.7 (± 6.7) | 44.5 (± 4.1) | 80.0 (± 1.4) | 59.1 (± 7.0) | 75.3 (± 2.9) | 72.8 (± 3.6) |

Table 5: Average classification accuracy (± standard deviation) on the 6 classification datasets containing node-labeled graphs.

| | Methods | DATASETS | | | | | |
| --- | --- | --- | --- | --- | --- | --- | --- |
| | | IMDB BINARY | IMDB MULTI | REDDIT BINARY | REDDIT MULTI-5K | REDDIT MULTI-12K | COLLAB |
| Kernels | VH | 50.0 (± 0.0) | 33.3 (± 0.0) | 50.0 (± 0.0) | 20.0 (± 0.0) | 21.7 (± 1.5) | 52.0 (± 0.1) |
| | RW | 64.1 (± 4.5) | 44.6 (± 4.1) | TIMEOUT | TIMEOUT | TIMEOUT | 68.0 (± 1.7) |
| | SP | 58.2 (± 4.7) | 39.2 (± 2.3) | 81.7 (± 2.5) | 47.9 (± 1.9) | TIMEOUT | 58.8 (± 1.2) |
| | GR | 66.1 (± 2.7) | 39.5 (± 2.7) | 76.1 (± 2.6) | 34.7 (± 2.0) | 23.0 (± 1.4) | 73.0 (± 2.0) |
| | WL-VH | 70.7 (± 6.8) | 51.3 (± 4.4) | 67.8 (± 3.5) | 50.5 (± 1.6) | 38.7 (± 1.7) | 78.3 (± 2.1) |
| | WL-SP | 58.2 (± 4.7) | 39.2 (± 2.3) | TIMEOUT | TIMEOUT | TIMEOUT | 58.8 (± 1.2) |
| | WL-PM | 73.6 (± 3.4) | 49.1 (± 5.5) | OUT-OF-MEM | OUT-OF-MEM | OUT-OF-MEM | OUT-OF-MEM |
| | WL-OA | 72.6 (± 5.5) | 51.1 (± 4.3) | 89.0 (± 1.3) | 54.0 (± 1.2) | TIMEOUT | 80.5 (± 2.0) |
| | NH | 71.6 (± 4.5) | 50.5 (± 5.0) | 81.2 (± 2.0) | 49.9 (± 2.4) | 39.6 (± 1.4) | 81.1 (± 2.4) |
| | NSPDK | 67.4 (± 3.3) | 44.6 (± 3.8) | TIMEOUT | TIMEOUT | TIMEOUT | TIMEOUT |
| | Lo-$\vartheta$ | 51.0 (± 4.2) | 39.8 (± 2.6) | TIMEOUT | TIMEOUT | TIMEOUT | TIMEOUT |
| | SVM-$\vartheta$ | 52.3 (± 4.0) | 39.5 (± 2.7) | 74.8 (± 2.6) | 31.4 (± 1.1) | 22.9 (± 0.9) | 52.0 (± 0.1) |
| | ODD-STh | 65.0 (± 4.0) | 46.7 (± 3.4) | 52.1 (± 3.2) | 43.1 (± 1.8) | 30.0 (± 1.6) | 52.0 (± 0.1) |
| | PM | 66.3 (± 4.2) | 46.1 (± 3.8) | 86.5 (± 2.1) | 48.3 (± 2.5) | 41.1 (± 0.6) | 74.0 (± 2.4) |
| | GH | 59.4 (± 3.4) | 39.5 (± 2.6) | TIMEOUT | TIMEOUT | TIMEOUT | 60.0 (± 1.4) |
| | SM | TIMEOUT | TIMEOUT | OUT-OF-MEM | OUT-OF-MEM | OUT-OF-MEM | TIMEOUT |
| | PK | 51.7 (± 3.7) | 34.5 (± 3.0) | 63.9 (± 3.0) | 34.9 (± 1.7) | 23.9 (± 1.2) | 57.0 (± 1.2) |
| | ML | 69.9 (± 4.8) | 47.7 (± 3.2) | 89.4 (± 2.1) | 35.4 (± 2.0) | OUT-OF-MEM | 75.6 (± 1.6) |
| | CORE-WL-VH | 73.5 (± 6.1) | 51.7 (± 4.1) | 73.0 (± 4.5) | 51.1 (± 1.6) | 40.2 (± 1.8) | 84.5 (± 2.0) |
| | CORE-SP | 68.5 (± 3.9) | 51.0 (± 3.5) | 91.0 (± 1.8) | TIMEOUT | OUT-OF-MEM | TIMEOUT |
| | Diffusion | 72.3 (± 1.8) | 50.8 (± 2.0) | 85.6 (± 1.2) | 49.5 (± 0.4) | 35.9 (± 0.7) | 75.1 (± 0.2) |
| | GRF | 68.9 (± 1.9) | 48.7 (± 2.4) | 84.0 (± 1.1) | 48.6 (± 0.7) | 35.1 (± 0.9) | 70.8 (± 0.6) |
| | GRF++ | 71.2 (± 2.1) | 49.1 (± 2.9) | 84.4 (± 1.5) | 49.2 (± 0.9) | 36.8 (± 0.9) | 73.6 (± 0.9) |
| GNNs | DGCNN | 69.2 (± 3.0) | 45.6 (± 3.4) | 87.8 (± 2.5) | 49.2 (± 1.2) | 43.9 (± 1.0) | 71.2 (± 1.9) |
| | GraphSAGE | 68.8 (± 4.5) | 47.6 (± 3.5) | 84.3 (± 1.9) | 50.0 (± 1.3) | 43.5 (± 1.0) | 73.9 (± 1.7) |
| | DiffPool | 68.4 (± 3.3) | 45.6 (± 3.4) | 89.1 (± 1.6) | 53.8 (± 1.4) | 44.4 (± 1.4) | 68.9 (± 2.0) |
| | GIN | 71.2 (± 3.9) | 48.5 (± 3.3) | 89.9 (± 1.9) | 56.1 (± 1.7) | 48.3 (± 1.6) | 75.6 (± 2.3) |

Table 6: Average classification accuracy (± standard deviation) on the 6 classification datasets containing unlabeled graphs.

Table 7: Final Ranking of Methods on the Unlabeled Datasets (Lower Average Rank is Better).

| Rank | Method | Average Rank |
|------|--------|--------------|
| 1 | GIN | 4.50 |
| 2 | CORE-WL-VH | 5.00 |
| 3 | NH | 6.67 |
| 4 | Diffusion | 7.00 |
| 5 | WL-OA | 7.17 |
| 6 | WL-VH | 7.33 |
| 7 | GraphSAGE | 8.83 |
| 7 | GRF++ (ours) | 8.83 |
| 7 | DiffPool | 8.83 |
| 10 | DGCNN | 9.50 |
| 11 | PM | 10.33 |
| 12 | GRF | 11.00 |
| 13 | ML | 11.83 |
| 14 | GR | 15.83 |
| 15 | ODD-STh | 16.33 |
| 16 | CORE-SP | 16.50 |
| 17 | SVM-$\vartheta$ | 19.17 |
| 17 | PK | 19.17 |
| 19 | SP | 19.33 |
| 20 | WL-PM | 19.50 |
| 21 | VH | 21.50 |
| 22 | RW | 22.17 |
| 23 | GH | 23.17 |
| 24 | NSPDK | 23.50 |
| 25 | WL-SP | 24.17 |
| 26 | Lo-$\vartheta$ | 25.33 |
| 27 | SM | 27.00 |

### B.3 Node Clustering

We follow the same setup as (Reid et al., 2024b) except that the number of clusters is based upon the actual number of different classes. Thus we use $p_{halt} = 0.1$, $m = 16$. To get details of the dataset, please see Ivashkin & Chebotarev (2016). In table 9, we compare GRF++ with 5 other methods : Louvain, Spectral, GRF, KCenters and Propagation. Our method is competitive across all datasets and outperforms all methods on 3 out of 5 datasets.

### B.4 Vertex Normal Prediction Experiments

In this sub-section, we present implementation details for vertex normal prediction experiments. All the experiments are run on free Google Colab with 12Gb of RAM.

For this task, we choose 40 meshes for 3D-printed objects of varying sizes from the Thingi10K dataset. Following (Choromanski et al., 2024), we choose the following meshes corresponding to the ids given by :

```
[60246, 85580, 40179, 964933, 1624039, 91657, 79183, 82407, 40172, 65414,
90431, 74449, 73464, 230349, 40171, 61193, 77938, 375276, 39463, 110793,
368622, 37326, 42435, 1514901, 65282, 116878, 550964, 409624, 101902,
73410, 87602, 255172, 98480, 57140, 285606, 96123, 203289, 87601, 409629,
37384, 57084]
```

We do a small search for the width of the kernel $\sigma \in \{.5, .6, .8\}$ for the baseline runs. For both GRF and GRF++, the number of walks are chosen from the subset $\{4, 8, 16\}$ and the halting probability of the walk is .1. The degree of GRF++ is chosen to be 2.

Table 8: Cosine Similarity for Meshes. GRF++ matches the performance of the baseline kernel (BF). GRF++r reuses the same walk and still outperform GRF.

| MESH SIZE | 64 | 99 | 146 | 148 | 155 | 182 | 222 | 246 | 290 | 313 |
|---|---|---|---|---|---|---|---|---|---|---|
| BF | 0.4255 | 0.7675 | 0.9424 | 0.4325 | 0.7095 | 0.9654 | 0.8715 | 0.7464 | 0.8895 | 0.5514 |
| GRF | 0.3083 | 0.6786 | 0.9348 | 0.3813 | 0.6831 | 0.9466 | 0.8569 | 0.6722 | 0.8651 | 0.5309 |
| GRF++ | 0.3889 | 0.7163 | 0.9367 | 0.4434 | 0.6892 | 0.9611 | 0.8684 | 0.7377 | 0.8679 | 0.5432 |
| GRF++r | 0.3905 | 0.7163 | 0.9327 | 0.4435 | 0.6867 | 0.9564 | 0.8510 | 0.6878 | 0.8464 | 0.5178 |

| MESH SIZE | 362 | 482 | 502 | 518 | 614 | 639 | 777 | 942 | 992 | 1012 |
|---|---|---|---|---|---|---|---|---|---|---|
| BF | 0.5884 | 0.9830 | 0.8881 | 0.4956 | 0.9172 | 0.8958 | 0.8022 | 0.8559 | 0.7206 | 0.9366 |
| GRF | 0.5751 | 0.9737 | 0.8673 | 0.4486 | 0.8866 | 0.8739 | 0.7812 | 0.8369 | 0.6967 | 0.9136 |
| GRF++ | 0.5821 | 0.9807 | 0.8843 | 0.4830 | 0.9084 | 0.8914 | 0.8039 | 0.8496 | 0.7144 | 0.9234 |
| GRF++r | 0.5688 | 0.9775 | 0.8772 | 0.4734 | 0.8982 | 0.8866 | 0.8019 | 0.8406 | 0.7085 | 0.9200 |

| MESH SIZE | 1094 | 1192 | 1849 | 2599 | 2626 | 2996 | 3072 | 3559 | 3715 | 4025 |
|---|---|---|---|---|---|---|---|---|---|---|
| BF | 0.9236 | 0.8297 | 0.9265 | 0.4987 | 0.8927 | 0.9326 | 0.4796 | 0.9356 | 0.9619 | 0.9669 |
| GRF | 0.9001 | 0.7987 | 0.9101 | 0.4065 | 0.8778 | 0.9171 | 0.4637 | 0.9208 | 0.9508 | 0.9588 |
| GRF++ | 0.9170 | 0.8167 | 0.9202 | 0.4615 | 0.8820 | 0.9277 | 0.4731 | 0.9293 | 0.9546 | 0.9646 |
| GRF++r | 0.9098 | 0.8134 | 0.9146 | 0.4209 | 0.8730 | 0.9210 | 0.4606 | 0.9281 | 0.9561 | 0.9620 |

| MESH SIZE | 5155 | 5985 | 6577 | 6911 | 7386 | 7953 | 8011 | 8261 | 8449 | 8800 |
|---|---|---|---|---|---|---|---|---|---|---|
| BF | 0.9011 | 0.9194 | 0.9622 | 0.9769 | 0.9437 | 0.9460 | 0.9382 | 0.9196 | 0.9276 | 0.9836 |
| GRF | 0.8833 | 0.9091 | 0.9525 | 0.9682 | 0.9308 | 0.9383 | 0.9233 | 0.9050 | 0.9139 | 0.9778 |
| GRF++ | 0.8931 | 0.9154 | 0.9599 | 0.9751 | 0.9374 | 0.9429 | 0.9321 | 0.9145 | 0.9205 | 0.9820 |
| GRF++r | 0.8896 | 0.9129 | 0.9561 | 0.9701 | 0.9348 | 0.9410 | 0.9269 | 0.9095 | 0.9160 | 0.9805 |

Table 9: Node Clustering: Comparing GRFs++ against various clustering algorithms. Our method outperforms all other baselines on 3 out of 5 datasets.

| Name | # Nodes | # clusters | Propagation | KCenters | Spectral | Louvain | GRF | GRF++[d=2] |
|---|---|---|---|---|---|---|---|---|
| Karate | 34 | 2 | 0.4011 | 0.2585 | 0.2585 | 0.4367 | 0.2995 | **0.2585** |
| Dolphins | 62 | 2 | 0.0936 | 0.0323 | 0.0408 | 0.3432 | 0.0635 | **0.0323** |
| Polbooks | 105 | 3 | **0.0621** | 0.0621 | 0.0692 | 0.1154 | 0.1060 | 0.1033 |
| Football | 115 | 12 | 0.1986 | 0.0477 | 0.0551 | **0.0174** | 0.0731 | 0.0362 |
| Databases | 1006 | 6 | 0.4762 | 0.4918 | 0.3214 | 0.5298 | 0.3528 | **0.3001** |
| Eurosis | 1272 | 13 | 0.7418 | 0.1891 | 0.1295 | 0.1599 | 0.2248 | **0.1304** |

## B.5 VISION TRANSFORMER ATTENTION MASKING

In this experiment, we investigate the utility of GRFs++ for introducing inductive biases into the self-attention mechanism of Vision Transformers (ViTs). We treat the input image patches as nodes in a regular 2D grid graph (lattice), where edges connect spatially adjacent patches. We employ GRFs++ (with degree $l = 2$) to efficiently approximate the diffusion kernel matrix $K_{\text{diff}}$ on this grid graph. This approximate kernel is subsequently normalized and utilized as a soft mask $M$, which is added to the standard attention logic:

$$\text{Attention}(Q, K, V) = \text{softmax}\left( \frac{QK^\top}{\sqrt{d}} + \lambda M \right) V \qquad (19)$$

We choose $\lambda = 0.5$ in our setup. Our empirical results on ImageNet show that the ViT-B-16 model with GRF++ masking achieves a top-1 accuracy of **80.71**%, surpassing the baseline ViT-B-16 performance of 80.16%.

## B.6 GRF++ BEYOND DIFFUSION KERNELS

In this section, we showcase the ability of GRFs++ to efficiently approximate graph node kernels beyond the diffusion kernel. We consider the following kernels: (1) $\mathbf{K}(\mathbf{W}) = (\mathbf{I} - \mathbf{W})^{-4}$, (2) $\mathbf{K}(\mathbf{W}) = (\mathbf{I} - \mathbf{W}^2)^{-4}$, (3) $\mathbf{K}(\mathbf{W}) = \exp(\mathbf{W}^2)$, with the corresponding modulation functions being given by Taylor-series expansions of the following matrix functions: (1) $\mathbf{W} \to (\mathbf{I} - \mathbf{W})^{-1}$, (2) $\mathbf{W} \to (\mathbf{I} - \mathbf{W}^2)^{-1}$, $\mathbf{W} \to \exp(\frac{\mathbf{W}^2}{4})$. The results are presented in Fig. 10,11,12. GRFs++ outperform regular GRFs for all three kernels.

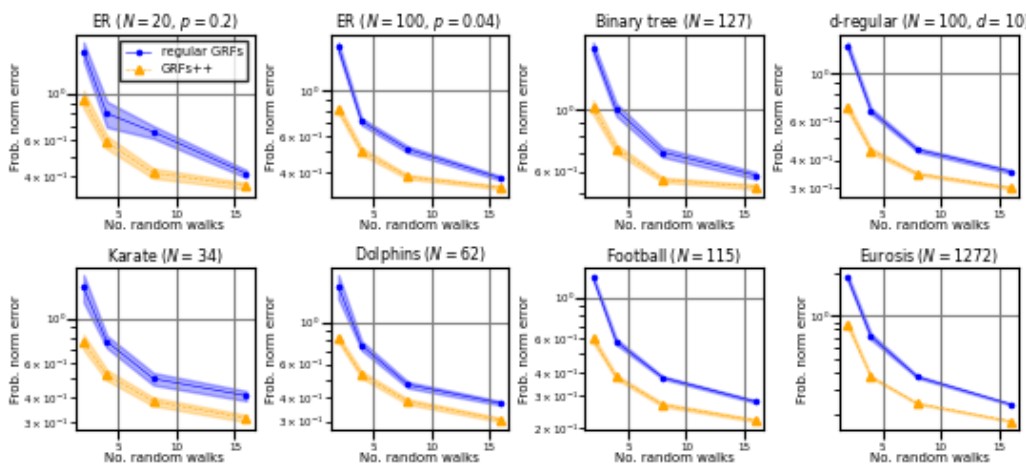

Figure 10: Experiment analogous to this from Fig. 2, but with a fixed degree $l = 2$ used in GRfs++ and graph kernel of the form $\mathbf{K}(\mathbf{W}) = (\mathbf{I} - \mathbf{W})^{-4}$.

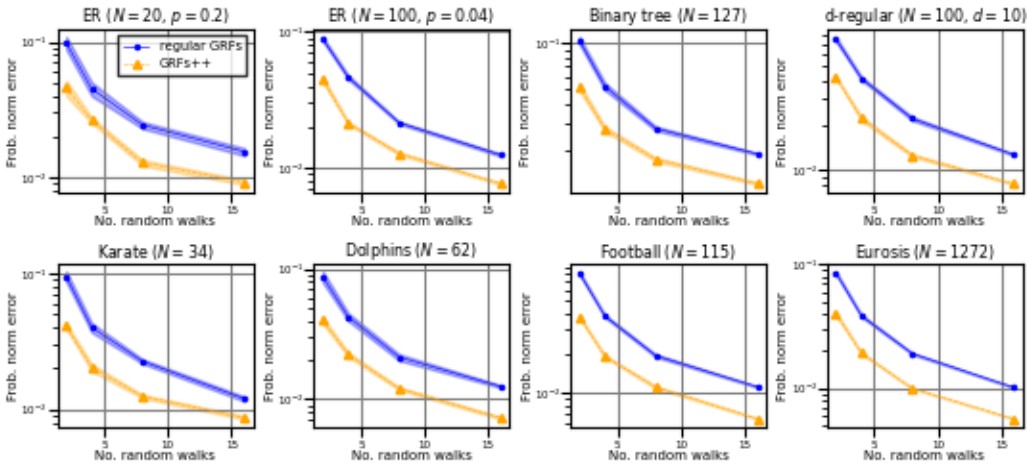

Figure 11: Experiment analogous to this from Fig. 2, but with a fixed degree $l = 2$ used in GRfs++ and graph kernel of the form $\mathbf{K}(\mathbf{W}) = (\mathbf{I} - \mathbf{W}^2)^{-4}$.

### B.7 CODE

The code is available at https://anonymous.4open.science/r/super_random_features_graphs-2782/README.md

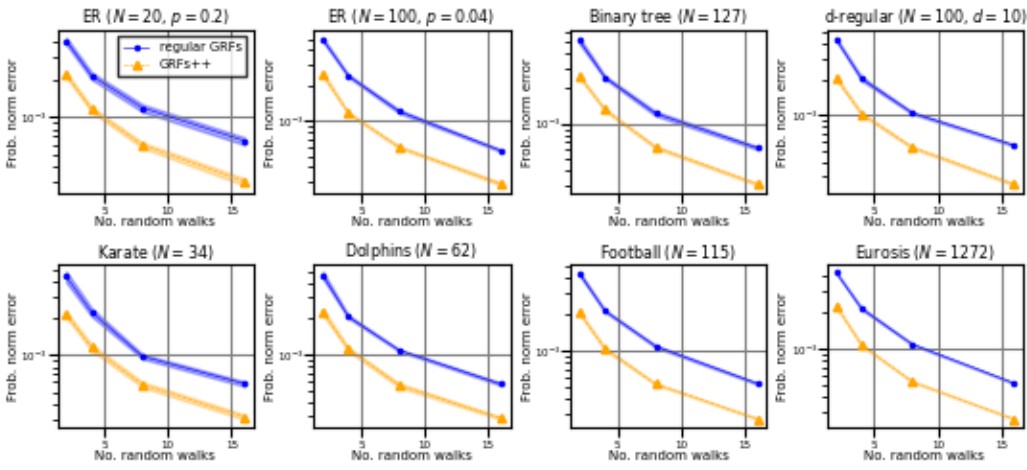

Figure 12: Experiment analogous to this from Fig. 2, but with a fixed degree $l = 2$ used in GRfs++ and graph kernel of the form $\mathbf{K}(\mathbf{W}) = \exp(\mathbf{W}^2)$.

