# OpenReview forum: "Computationally Efficient Graph Modelling with Refined Graph Random Features"
_ICLR.cc/2026/Conference — Submitted to ICLR 2026_

### Official Review · Reviewer_aor6 · 2025-10-20

**Soundness:** 4
**Presentation:** 2
**Contribution:** 2
**Rating:** 4
**Confidence:** 2

**Summary:**

The authors study the problem of graph random features as efficient and accurate graph kernels. Kernels are similarity functions defining relationships between graph nodes (or graphs themselves) of which the exact computation tends to be expensive. The idea of a graph random feature is to consider random walks in the graphs, where the length and number of such walks allow to make a statement about graph similarity. The key idea of this paper, is to improve the standard random walk procedure with a stitching operation and a termination criterion. This allows a improved approximation of the kernel, especially for distant nodes (demonstrated in a theoretical analysis, as well as experimentally), and reduces the computational cost.

**Strengths:**

- The walk-stitching technique sounds very logical and straightforward, to the point that one may wonder why nobody has considered to this before. (As an outsider to this field, I wonder if indeed nobody has.)
- The technical contributions come with a theoretical analysis.
- The experiments are comprehensive, on both synthetic and real data, and with downstream applications.

**Weaknesses:**

Major:
- While the advantages of GRF++ are clear, the need for GRF(++) should be better motivated in the Introduction. It is unclear to which extend GRF are needed and used in the first place. This makes the importance of the contribution hard to assess.
- The paper is sometimes hard to follow due to its structure. For instance, there is no clear narrative arc in Sec. 2.2.3. Same remark for Sec 3.
- The stitching time shown in Fig. 5 is unclear. Why does standard GRF (which does not use stitching) have a non-zero stitching time? How does the stitching time reduce while the stitching degree increases? (cf. Q2).
- Results in Appendix do not look as good as in the main paper, e.g., in Fig. 7 and 8 where the Frobenius norm error for higher degrees gets higher than for lower degrees, which seems to contradict Theorem 3.3. (cf. Q3)
- There is no discussion on how to choose the degree l. (cf Q1)

Minor:
- Figure text size too small (e.g., Fig. 3, 4 and 5)
- Algorithm 2 should at least be fully written in Appendix.
- Undefined notation/typo "R" l. 039
- Duplicated ref (Beutel et al., 2015a and 2015b)
- References to datasets karate, dolphin, football and eurosis seem to be missing.
- A large part of the references are unrelated (about graph modeling in anomaly detection, recommander system, and computational biology).
- Fig. 6 should be a table.
- The metric presented in Tab. 1 is not indicated in it nor in its label.

**Questions:**

Q1: It seems that a higher degree l leads to a better approximation (Theorem 3.3.) and a lower computation time thanks to the parallelization and the reuse of the sub-walks (experimental results). Is there a point at which increasing the degree will result in less efficient computation due to stitching time? If so, how should l be chosen? If not, why should l not be chosen as high as the expected longest path? And why was l=2 chosen in the downstream applications?
Q2: Please clarify the meaning of stitching time in Fig. 5.
Q3: How do you intreprete the "crossing lines" in Fig. 7 and 8, which seem to contradict Theorem 3.3?

---

> ### Author Response · Authors · 2025-11-21
> **Answers to Reviewer's comment: part I**
>
> We would like to sincerely thank the Reviewer for the feedback. We address all the questions below.
>
> **The importance of GRFs:**
>
> Thank you very much for the comment. We would like to emphasize that the goal of this paper is not to provide a strong use-case for GRFs, but to improve this well-established class of graph algorithms. Having said that,  following Reviewer’s comments, we have added several additional experiments, all demonstrating that GRFs++ are competitive and often better than other graph modeling techniques, used on a regular basis, and therefore important in several practical applications.
>
> $\textit{Clustering Results:}$
>
> We compared GRFs++ in graph clustering applications against various clustering algorithms like Propagation, KCenters, Spectral and Louvain. Our method outperforms all other baselines on 3 out of 5 datasets (see: table below).
>
> | Name       | # Nodes | # Clusters | Propagation | KCenters | Spectral | Louvain | GRF    | GRF++ [d=2] |
> |------------|---------|------------|-------------|----------|----------|---------|--------|-------------|
> | Karate     | 34      | 2          | 0.4011      | 0.2585   | 0.2585   | 0.4367  | 0.2995 | **0.2585**  |
> | Dolphins   | 62      | 2          | 0.0936      | 0.0323   | 0.0408   | 0.3432  | 0.0635 | **0.0323**  |
> | Polbooks   | 105     | 3          | **0.0621**  | 0.0621   | 0.0692   | 0.1154  | 0.1060 | 0.1033      |
> | Football   | 115     | 12         | 0.1986      | 0.0477   | 0.0551   | **0.0174** | 0.0731 | 0.0362   |
> | Databases  | 1006    | 6          | 0.4762      | 0.4918   | 0.3214   | 0.5298  | 0.3528 | **0.3001**  |
> | Eurosis    | 1272    | 13         | 0.7418      | 0.1891   | 0.1295   | 0.1599  | 0.2248 | **0.1304**  |
>
>
> $\textit{Node Classification Results:}$
>
>
> We apply the diffusion kernel in GCN similarly as in [1]. GRF++ matches or outperforms a regular GCN diffusion kernel variant (see: table below). The accuracy results are slightly lower than that of the baseline GCN due to possible oversmoothing effects.
>
> | DATASET  | DIFFUSION-GCN | GRF++ | GRF |
> |----------|-----------|-------|-----|
> | CORA     | 78.6 ± 0.9 | 79.8 ± 0.7 | 78.1 ± 0.6 |
> | CITESEER | 68.9 ± 0.3 | 69.7 ± 0.5 | 68.2 ± 0.7 |
> | PUBMED   | 77.9 ± 0.4 | 77.8 ± 0.3 | 76.8 ± 0.4 |
>
>
> [1] Adaptive Diffusion in Graph Neural Networks Zhao et al. Neurips 2021
>
>
> $\textit{ViT Results}:$
>
> We have integrated our GRF++ into ViT models, using the approximate graph kernel matrix (for the 2D-grid graph) given by GRFs++ as a mask to provide inductive bias to the attention mechanism. Our masked ViT improves the baseline ViT on ImageNet, achieving **80.71**% top-1 accuracy compared to 80.16%. We provide additional details in Section B.5.

---

> > ### Author Response · Authors · 2025-11-21
> > **Answers to Reviewer's comments: part II**
> >
> > **The importance of GRFs**:
> >
> > $\textit{Graph Classification Results:}$
> >
> > Finally, as requested by the Reviewer, we posit our results in the context of various kernel methods as well as GNNs. In particular, we compare our methods against **20** popular kernels: (1) Vertex Histogram kernel (VH), (2) Random Walk kernel (RW), (3)
> > Shortest Path kernel (SP), (4) Graphlet kernel (GR), (5) Weisfeiler-Lehman sub-tree kernel (WL-VH), (6) Weisfeiler-Lehman shortest path kernel (WL-SP), (7) Weisfeiler-Lehman
> > pyramid match kernel (WL-PM), (8) Weisfeiler-Lehman optimal assignment kernel (WLOA), (9) Neighborhood Hash kernel (NH), (10) Neighborhood subgraph pairwise distance
> > kernel (NSPDK), (11) Lov\'asz $\vartheta$ kernel (Lo-$\vartheta$), (12) SVM-$\vartheta$ kernel (SVM-$\vartheta$), (13) Ordered Decompositional DAGs with subtree kernel (ODD-STh), (14) Pyramid Match kernel (PM), (15) GraphHopper kernel (GH), (16) Subgraph Matching kernel (SM), (17) Propagation kernel (PK), (18) Multiscale Laplacian kernel (ML), (19) Core Weisfeiler-Lehman subtree
> > kernel (CORE-WL-VH), and (20) Core Shortest Path kernel (CORE-SP). Note that this kernel computes relationships between graphs whereas our GRF++ computes relationships between nodes in a given graph. Finally we also compare our methods against popular GNN models: (1) DGCNN [1], (2) GraphSAGE [2], (3) DiffPool [3], and (4) GIN [4]. All these baseline numbers are taken from [5]. Our results show that the diffusion kernel is competitive across all the datasets. Our GRF++ mechanism maintains similar performance to that of the baseline diffusion kernel while being more computationally efficient. Detailed results are presented in Tables 5 and 6 in the updated version of the paper, along with additional results on 2 more datasets : **Reddit-Multi-12k** and **Collab**. Below we summarize the results for graphs with unlabeled nodes. GRF++ based algorithm is one of the top-performing methods.
> >
> >
> > | Rank | Method        | Average Rank |
> > |------|--------------|--------------|
> > | 1    | GIN          | 4.50         |
> > | 2    | CORE-WL-VH   | 5.00         |
> > | 3    | NH           | 6.67         |
> > | 4    | Diffusion    | 7.00         |
> > | 5    | WL-OA        | 7.17         |
> > | 6    | WL-VH        | 7.33         |
> > | 7    | GraphSAGE    | 8.83         |
> > | 7    | GRF++ (ours)       | 8.83         |
> > | 7    | DiffPool     | 8.83         |
> > | 10   | DGCNN        | 9.50         |
> > | 11   | PM           | 10.33        |
> > | 12   | GRF          | 11.00        |
> > | 13   | ML           | 11.83        |
> > | 14   | GR           | 15.83        |
> > | 15   | ODD-STh      | 16.33        |
> > | 16   | CORE-SP      | 16.50        |
> > | 17   | SVM-θ        | 19.17        |
> > | 17   | PK           | 19.17        |
> > | 19   | SP           | 19.33        |
> > | 20   | WL-PM        | 19.50        |
> > | 21   | VH           | 21.50        |
> > | 22   | RW           | 22.17        |
> > | 23   | GH           | 23.17        |
> > | 24   | NSPDK        | 23.50        |
> > | 25   | WL-SP        | 24.17        |
> > | 26   | Lo-θ         | 25.33        |
> > | 27   | SM           | 27.00        |
> >
> >
> > [1] An End-to-End Deep Learning Architecture for Graph Classification. Zhang et al. AAAI 2018
> >
> > [2] Inductive representation learning on large graphs. Hamilton et al. NeurIPS 2017
> >
> > [3] Hierarchical graph representation learning with differentiable pooling. Ying et al. NeurIPS 2018
> >
> > [4] How Powerful are Graph Neural Networks? Xu et al. ICLR 2019
> >
> > [5] Graph Kernels: A Survey. Nikolentzos et al. JAIR 2021

---

> > > ### Author Response · Authors · 2025-11-21
> > > **Answers to Reviewer's comments: part III**
> > >
> > > **Sec. 2.2.3 and Sec. 3:**
> > >
> > > Thank you very much for the comment. We will further polish the narrative of those sections in the camera-ready version of the paper.
> > >
> > > **Stitching time in Fig. 5:**
> > >
> > > Thank you for your question. We refer to “stitching time” in Fig. 5 as the time required to make the matrix-matrix multiplications with the already computed sequence of factor matrices. If no stitching technique is applied, then this sequence contains only two matrices, but the corresponding calculations still need to be conducted. We agree that the name “stitching time” might be confusing and thus we will use another name in the final version of the paper.
> > >
> > > **Results in the Appendix:**
> > >
> > > As mentioned before, we have added several new experimental results in the Appendix, all confirming strong performance of GRFs++.
> > >
> > > **The choice of the degree l:**
> > >
> > > Thank you for the great questions. Stitching provides a tradeoff between parallel computation for the “chunks of random walks” and increased number of matrix-matrix multiplication. The optimal choice of l depends in practice on the corresponding wall-clock times for both. We noted experimentally that a low-degree l=2 variant already provides significant computational gains in practice, while adding a minimal number of extra matrix-matrix multiplications. That is why it is used in most of the experiments in this paper. We would like to note though that for instance in Fig. 2 we use a large portfolio of degrees for the accuracy tests.
> > >
> > > **Cross-lines in Fig.7 and Fig. 8:**
> > >
> > > A few cross-lines in those figures correspond to the fact that presented errors are empirical (with a very small number of random walks being used). Increasing the number of random walks effectively eliminates all the crossings, because of the presented theoretical results.
> > >
> > > **Minor:**
> > >
> > > Thank you very much for the comment. We have already fixed some typos (e.g. duplicate references) and will make additional suggested edits in the camera-ready version of the paper.

---

### Official Review · Reviewer_B661 · 2025-10-26

**Soundness:** 2
**Presentation:** 3
**Contribution:** 2
**Rating:** 4
**Confidence:** 4

**Summary:**

The paper proposes an efficient extension to the graph kernel framework g-GRF (Reid et al., 2024b) by introducing a random walk-stitching technique that combines shorter walks to approximate longer ones. This approach aims to reduce the computational cost of graph node kernel computation while maintaining representational power. The authors provide a theoretical analysis of the proposed method, building upon the foundations of the original g-GRF framework. Empirical evaluations are conducted on graph classification and clustering tasks, demonstrating improved efficiency and competitive performance.

**Strengths:**

- The paper is generally well-organized and clearly written. The connection to the previous g-GRF framework is articulated in a way that makes the contribution easy to follow.
- The introduced random walk-stitching mechanism offers a novel perspective on constructing longer random walks from shorter ones.
- The paper presents a theoretical analysis grounded in the prior g-GRF formalism.

**Weaknesses:**

- The paper provides a limited novelty. It extends the previously introduced g-GRF framework (Reid et al., 2024b) with a relatively incremental modification.
- The experimental evaluation is limited in baseline coverage. For graph classification, a baseline and g-GRF are included; similarly, for clustering, no baseline beyond GRF is considered. Standard deviations are not reported
- Although the method claims improved computational efficiency, the practical benefits are not convincingly demonstrated. Evaluations are conducted on small graphs.

**Questions:**

- The evaluation currently relies on relatively small datasets and mainly compares against g-GRF variants. Would the proposed method maintain its efficiency and accuracy advantages on larger or more complex datasets?

- The empirical results show that GRF++ outperforms GRF in most classification settings. Could the authors provide an intuition or theoretical justification for why the proposed random walk-stitching mechanism leads to such gains? Including or at least discussing scalability experiments on larger graphs would strengthen the empirical claims. Similarly, adding comparisons to more recent GNN-based baselines would better contextualize the contribution within current literature.
- For the clustering experiments, it would be useful to include baseline methods beyond GRF and to report standard deviation or confidence interval values over multiple runs.

**Additional comments:**
- In Line 107, the Eq 2 cannot be convergent for any arbitrary $(\alpha_k)_{k=0}$, for example, for divergent $\alpha_k$ values.

---

> ### Author Response · Authors · 2025-11-21
> **Answers to Reviewer's questions - part I**
>
> We would like to sincerely thank the Reviewer for the feedback. We address all the questions below.
>
> **Novelty:**
>
> Thank you for the comment. We disagree with the Reviewer that the novelty is limited. For instance, the fact that stitching mechanism provides MSE improvements for an arbitrary graph kernel from the considered family is a nontrivial theoretical result. The paper opens new important research directions on GRFs, in particular regarding designing optimal termination strategies. This is yet another nontrivial theoretical question, addressed empirically by GRFs++ via the Poisson distribution method. Most importantly, in its current form GRFs++ already provide significant improvements, as compared to regular GRFs in **practical applications**, by bypassing a notoriously difficult problem of running sequentially long non-parallelizable random walks. This was one of the main technical obstacles to make the GRF method more accelerator-friendly and it is resolved here.

---

> > ### Author Response · Authors · 2025-11-21
> > **Answers to Reviewer's questions - part II**
> >
> > **Limited Experimental Evaluation:**
> >
> > Following Reviewer’s comments, we have added several additional experiments, all demonstrating that GRFs++ are competitive and often better than other graph modeling techniques, used on a regular basis.
> >
> > $\textit{Clustering Results:}$
> >
> > We compared GRFs++ in graph clustering applications against various clustering algorithms like Propagation, KCenters, Spectral and Louvain. Our method outperforms all other baselines on 3 out of 5 datasets (see: table below).
> >
> > | Name       | # Nodes | # Clusters | Propagation | KCenters | Spectral | Louvain | GRF    | GRF++ [d=2] |
> > |------------|---------|------------|-------------|----------|----------|---------|--------|-------------|
> > | Karate     | 34      | 2          | 0.4011      | 0.2585   | 0.2585   | 0.4367  | 0.2995 | **0.2585**  |
> > | Dolphins   | 62      | 2          | 0.0936      | 0.0323   | 0.0408   | 0.3432  | 0.0635 | **0.0323**  |
> > | Polbooks   | 105     | 3          | **0.0621**  | 0.0621   | 0.0692   | 0.1154  | 0.1060 | 0.1033      |
> > | Football   | 115     | 12         | 0.1986      | 0.0477   | 0.0551   | **0.0174** | 0.0731 | 0.0362   |
> > | Databases  | 1006    | 6          | 0.4762      | 0.4918   | 0.3214   | 0.5298  | 0.3528 | **0.3001**  |
> > | Eurosis    | 1272    | 13         | 0.7418      | 0.1891   | 0.1295   | 0.1599  | 0.2248 | **0.1304**  |
> >
> >
> > $\textit{ViT Results:}$
> >
> > We have integrated our GRF++ into ViT models, using the approximate graph kernel matrix (for the 2D-grid graph) given by GRFs++ as a mask to provide inductive bias to the attention mechanism. Our masked ViT improves the baseline ViT on ImageNet, achieving **80.71**% top-1 accuracy compared to 80.16%. We provide additional details in Section B.5.

---

> > > ### Author Response · Authors · 2025-11-21
> > > **Answers to Reviewer's questions: part III**
> > >
> > > **Limited Experimental Evaluation:**
> > >
> > > $\textit{Graph Classification Results}:$
> > >
> > > Finally, as requested by the Reviewer, we posit our results in the context of various kernel methods as well as GNNs. In particular, we compare our methods against **20** popular kernels: (1) Vertex Histogram kernel (VH), (2) Random Walk kernel (RW), (3)
> > > Shortest Path kernel (SP), (4) Graphlet kernel (GR), (5) Weisfeiler-Lehman sub-tree kernel (WL-VH), (6) Weisfeiler-Lehman shortest path kernel (WL-SP), (7) Weisfeiler-Lehman
> > > pyramid match kernel (WL-PM), (8) Weisfeiler-Lehman optimal assignment kernel (WLOA), (9) Neighborhood Hash kernel (NH), (10) Neighborhood subgraph pairwise distance
> > > kernel (NSPDK), (11) Lov\'asz $\vartheta$ kernel (Lo-$\vartheta$), (12) SVM-$\vartheta$ kernel (SVM-$\vartheta$), (13) Ordered Decompositional DAGs with subtree kernel (ODD-STh), (14) Pyramid Match kernel (PM), (15) GraphHopper kernel (GH), (16) Subgraph Matching kernel (SM), (17) Propagation kernel (PK), (18) Multiscale Laplacian kernel (ML), (19) Core Weisfeiler-Lehman subtree
> > > kernel (CORE-WL-VH), and (20) Core Shortest Path kernel (CORE-SP). Note that this kernel computes relationships between graphs whereas our GRF++ computes relationships between nodes in a given graph. Finally we also compare our methods against popular GNN models: (1) DGCNN [1], (2) GraphSAGE [2], (3) DiffPool [3], and (4) GIN [4]. All these baseline numbers are taken from [5]. Our results show that the diffusion kernel is competitive across all the datasets. Our GRF++ mechanism maintains similar performance to that of the baseline diffusion kernel while being more computationally efficient. Detailed results are presented in Tables 5 and 6 in the updated version of the paper, along with additional results on 2 more datasets : **Reddit-Multi-12k** and **Collab**. Below we summarize the results for graphs with unlabeled nodes. GRF++ based algorithm is one of the top-performing methods.
> > >
> > >
> > > | Rank | Method        | Average Rank |
> > > |------|--------------|--------------|
> > > | 1    | GIN          | 4.50         |
> > > | 2    | CORE-WL-VH   | 5.00         |
> > > | 3    | NH           | 6.67         |
> > > | 4    | Diffusion    | 7.00         |
> > > | 5    | WL-OA        | 7.17         |
> > > | 6    | WL-VH        | 7.33         |
> > > | 7    | GraphSAGE    | 8.83         |
> > > | 7    | GRF++ (ours)       | 8.83         |
> > > | 7    | DiffPool     | 8.83         |
> > > | 10   | DGCNN        | 9.50         |
> > > | 11   | PM           | 10.33        |
> > > | 12   | GRF          | 11.00        |
> > > | 13   | ML           | 11.83        |
> > > | 14   | GR           | 15.83        |
> > > | 15   | ODD-STh      | 16.33        |
> > > | 16   | CORE-SP      | 16.50        |
> > > | 17   | SVM-θ        | 19.17        |
> > > | 17   | PK           | 19.17        |
> > > | 19   | SP           | 19.33        |
> > > | 20   | WL-PM        | 19.50        |
> > > | 21   | VH           | 21.50        |
> > > | 22   | RW           | 22.17        |
> > > | 23   | GH           | 23.17        |
> > > | 24   | NSPDK        | 23.50        |
> > > | 25   | WL-SP        | 24.17        |
> > > | 26   | Lo-θ         | 25.33        |
> > > | 27   | SM           | 27.00        |
> > >
> > >
> > > [1] An End-to-End Deep Learning Architecture for Graph Classification. Zhang et al. AAAI 2018
> > >
> > > [2] Inductive representation learning on large graphs. Hamilton et al. NeurIPS 2017
> > >
> > > [3] Hierarchical graph representation learning with differentiable pooling. Ying et al. NeurIPS 2018
> > >
> > > [4] How Powerful are Graph Neural Networks? Xu et al. ICLR 2019
> > >
> > > [5] Graph Kernels: A Survey. Nikolentzos et al. JAIR 2021
> > >
> > >
> > > **Computational efficiency:**
> > >
> > > We highlight the speedups of GRF++ over the baseline brute force kernel method in the table below. A detailed plot showing our results over **40** meshes of various sizes are presented in Fig 7 in the updated version of the paper.
> > >
> > > | Mesh Size | BF Time (s) | GRF++ Time (s) | Speedup (BF / GRF++) |
> > > |-----------:|------------:|---------------:|----------------------:|
> > > | 2599       | 17.73       | 10.58          | **1.68× faster** |
> > > | 2996       | 19.61       | 11.31          | **1.73× faster** |
> > > | 3072       | 23.55       | 11.77          | **2.00× faster** |
> > > | 3715       | 37.67       | 17.48          | **2.15× faster** |
> > > | 5985       | 109.89      | 46.68          | **2.35× faster** |
> > > | 6577       | 146.51      | 62.77          | **2.33× faster** |
> > > | 7386       | 205.82      | 68.25          | **3.01× faster** |
> > > | 7953       | 252.66      | 82.19          | **3.07× faster** |
> > > | 8261       | 267.71      | 88.32          | **3.03× faster** |
> > > | 9603       | 417.65      | 128.92         | **3.24× faster** |
> > >
> > > Conclusion: At moderate mesh sizes (2K–4K), GRF++ gives ~1.7×–2.2× speedup while for larger meshes (6K–10K), GRF++ consistently delivers ~3× faster runtime. Thus GRF++ scales significantly better as mesh complexity increases.

---

> > > > ### Author Response · Authors · 2025-11-21
> > > > **Answers to Reviewer's questions: part IV**
> > > >
> > > > **Intuition regarding the improvements provided by GRFs++, as compared to GRFs, on the classification tasks:**
> > > >
> > > > GRFs++ lead to provably more accurate estimation of the underlying class of graph kernels than GRFs. The classification task results show that those graph kernels in practice are useful to solve these tasks. Thus methods approximating those kernels more closely are in general more effective.
> > > >
> > > > **Line 107:**
> > > >
> > > > Thank you for the comment. Yes, the Reviewer is right that the convergence is not achieved for an arbitrary sequence of coefficients $(\alpha_{k})$. Thank you for pointing out this typo. We fixed it in the updated version where we take bounded $(\alpha_{k})$. Of course all the presented theoretical and empirical results still hold.

---

> > > > > ### Comment · Reviewer_B661 · 2025-11-27
> > > > > **Thank you for your responses and for the additional experimental results**
> > > > >
> > > > > Thank you for your responses and for providing additional experimental results. Regarding the novelty of the work, my assessment remains unchanged. Concerning the theoretical contribution, Lemma 3.2 seems to be a direct adaptation of the well-known variance formula for the variable $X_1X_2$. Similarly, Lemma 2.1 closely follows the results presented in g-GRF (Reid et al., 2024b). While the paper states that any discrete distribution can be used, the experiments also only focus on the Poisson distribution, and the choice of the rate parameter is not discussed.
> > > > >
> > > > > For the clustering experiments, for instance, the spectral method performs on par or better than GRF++ (it also achieves the best performance on the Eurosis dataset). The datasets used in these evaluations are also very small. In the graph classification task, a relatively older GNN method, GraphSAGE, shows comparable performance, which makes it unclear to me why the proposed approach would be preferred in practical applications. Therefore, my overall evaluation remains unchanged

---

### Official Review · Reviewer_r6Zd · 2025-10-30

**Soundness:** 4
**Presentation:** 4
**Contribution:** 2
**Rating:** 4
**Confidence:** 4

**Summary:**

This paper proposes GRFs++, a method to approximate a large family of kernels between nodes including the d-regularized Laplacian, diffusion process and p-step random walk kernel. The proposed method is based on GRFs, a method that approximates the kernels matrix as a product of two low rank matrices whose elements are obtained via random walks. GRFs++ deals with the limitations of GRFs, in particular its inability to model relationships between distant nodes and its walk termination mechanism. Experimental results show that GRFs++ are more accurate and more efficient than GRFs.

**Strengths:**

- The proposed GRFs++ method is well-motivated, as it addresses certain limitations of the prior GRFs approach. Combining shorter walks to construct longer walks is an interesting and reasonable idea.

- The method is supported by theoretical analysis. Among others, the authors show that increasing the walk-stitching degree in powers of 2 leads to improved approximation, which is an interesting result.

- GRFs++ demonstrates superior empirical performance compared to GRFs, achieving lower approximation error and better results in downstream tasks such as in graph classification and node clustering.

**Weaknesses:**

- My main concern with this paper is the significance of its contributions since the kernels between nodes that are approximated by the proposed GRFs++ are now rarely used in practice, having been largely replaced by GNNs that learn task-dependent features.

- In line with the previous point, it appears that the kernels that are approximated by GRFs++ are not empirically strong and thus might not be useful in practice. For example, the classification accuracies that are illustrated in Figure 6 are not considered state-of-the-art. On ENZYMES, NCI1 and REDDIT-MULTI, the accuracies are significantly lower than those of standard GNNs that are evaluated under the same protocol [1].

- No speed comparison is provided between the proposed approximation method and the approach that computes the exact kernel. I would expect that, even for graphs with 500 nodes, computing the exact kernel is not particularly computationally intensive. Such a comparison is missing from the paper.

- The proposed method is not compared against other types of approaches in the graph classification and node clustering tasks. For example, in the graph classification task, it could be compared against kernels between graphs and GNNs and in node clustering against spectral clustering, Louvain or some node embedding approach followed by k-means. More importantly, in terms of kernel approximation, it is only compared against GRFs. In my understanding, the well-known Nystrom method cannot be applied to this type of kernels. Is there any general approach that allows approximating this family of kernels?

[1] Errica, F., Podda, M., Bacciu, D., & Micheli, A. A fair comparison of graph neural networks for graph classification. In ICLR'20.

**Questions:**

Given that the proposed method approximates kernels between nodes, why isn't it evaluated in node classification tasks (Cora, Citeseer, etc.)?

---

> ### Author Response · Authors · 2025-11-21
> **Answers to Reviewer's questions - part I**
>
> We would like to sincerely thank the Reviewer for the feedback. We address all the questions below.
>
> **Significance of the results:**
>
> Thank you very much for the comments. We would like to emphasize that the goal of this paper is not to provide another SOTA GNN-method, but to improve a well-established class of GRF-based algorithms. GRFs++ provide consistent quality and speed gains, as thoroughly shown in the paper. Having said that,  following Reviewer’s comments, we have added several additional experiments, all demonstrating that GRFs++ are competitive and often better than other graph modeling techniques, used on a regular basis.
>
> $\textit{Clustering Results:}$
>
> We compared GRFs++ in graph clustering applications against various clustering algorithms like Propagation, KCenters, Spectral and Louvain. Our method outperforms all other baselines on 3 out of 5 datasets (see: table below).
>
> | Name       | # Nodes | # Clusters | Propagation | KCenters | Spectral | Louvain | GRF    | GRF++ [d=2] |
> |------------|---------|------------|-------------|----------|----------|---------|--------|-------------|
> | Karate     | 34      | 2          | 0.4011      | 0.2585   | 0.2585   | 0.4367  | 0.2995 | **0.2585**  |
> | Dolphins   | 62      | 2          | 0.0936      | 0.0323   | 0.0408   | 0.3432  | 0.0635 | **0.0323**  |
> | Polbooks   | 105     | 3          | **0.0621**  | 0.0621   | 0.0692   | 0.1154  | 0.1060 | 0.1033      |
> | Football   | 115     | 12         | 0.1986      | 0.0477   | 0.0551   | **0.0174** | 0.0731 | 0.0362   |
> | Databases  | 1006    | 6          | 0.4762      | 0.4918   | 0.3214   | 0.5298  | 0.3528 | **0.3001**  |
> | Eurosis    | 1272    | 13         | 0.7418      | 0.1891   | 0.1295   | 0.1599  | 0.2248 | **0.1304**  |
>
>
> $\textit{Node Classification Results:}$
>
>
> We apply the diffusion kernel in GCN similarly as in [1]. GRF++ matches or outperforms a regular GCN diffusion kernel variant (see: table below). The accuracy results are slightly lower than that of the baseline GCN due to possible oversmoothing effects.
>
> | DATASET  | DIFFUSION-GCN | GRF++ | GRF |
> |----------|-----------|-------|-----|
> | CORA     | 78.6 ± 0.9 | 79.8 ± 0.7 | 78.1 ± 0.6 |
> | CITESEER | 68.9 ± 0.3 | 69.7 ± 0.5 | 68.2 ± 0.7 |
> | PUBMED   | 77.9 ± 0.4 | 77.8 ± 0.3 | 76.8 ± 0.4 |
>
>
> [1] Adaptive Diffusion in Graph Neural Networks Zhao et al. Neurips 2021
>
>
> $\textit{ViT Results}:$
>
> We have integrated our GRF++ into ViT models, using the approximate graph kernel matrix (for the 2D-grid graph) given by GRFs++ as a mask to provide inductive bias to the attention mechanism. Our masked ViT improves the baseline ViT on ImageNet, achieving **80.71**% top-1 accuracy compared to 80.16%. We provide additional details in Section B.5.

---

> > ### Author Response · Authors · 2025-11-21
> > **Answers to Reviewer's questions - part II**
> >
> > **Significance of the results:**
> >
> > $\textit{Graph Classification Results:}$
> >
> > Finally, as requested by the Reviewer, we posit our results in the context of various kernel methods as well as GNNs. In particular, we compare our methods against **20** popular kernels: (1) Vertex Histogram kernel (VH), (2) Random Walk kernel (RW), (3)
> > Shortest Path kernel (SP), (4) Graphlet kernel (GR), (5) Weisfeiler-Lehman sub-tree kernel (WL-VH), (6) Weisfeiler-Lehman shortest path kernel (WL-SP), (7) Weisfeiler-Lehman
> > pyramid match kernel (WL-PM), (8) Weisfeiler-Lehman optimal assignment kernel (WLOA), (9) Neighborhood Hash kernel (NH), (10) Neighborhood subgraph pairwise distance
> > kernel (NSPDK), (11) Lov\'asz $\vartheta$ kernel (Lo-$\vartheta$), (12) SVM-$\vartheta$ kernel (SVM-$\vartheta$), (13) Ordered Decompositional DAGs with subtree kernel (ODD-STh), (14) Pyramid Match kernel (PM), (15) GraphHopper kernel (GH), (16) Subgraph Matching kernel (SM), (17) Propagation kernel (PK), (18) Multiscale Laplacian kernel (ML), (19) Core Weisfeiler-Lehman subtree
> > kernel (CORE-WL-VH), and (20) Core Shortest Path kernel (CORE-SP). Note that this kernel computes relationships between graphs whereas our GRF++ computes relationships between nodes in a given graph. Finally we also compare our methods against popular GNN models: (1) DGCNN [1], (2) GraphSAGE [2], (3) DiffPool [3], and (4) GIN [4]. All these baseline numbers are taken from [5]. Our results show that the diffusion kernel is competitive across all the datasets. Our GRF++ mechanism maintains similar performance to that of the baseline diffusion kernel while being more computationally efficient. Detailed results are presented in Tables 5 and 6 in the updated version of the paper, along with additional results on 2 more datasets : **Reddit-Multi-12k** and **Collab**. Below we summarize the results for graphs with unlabeled nodes. GRF++ based algorithm is one of the top-performing methods.
> >
> >
> > | Rank | Method        | Average Rank |
> > |------|--------------|--------------|
> > | 1    | GIN          | 4.50         |
> > | 2    | CORE-WL-VH   | 5.00         |
> > | 3    | NH           | 6.67         |
> > | 4    | Diffusion    | 7.00         |
> > | 5    | WL-OA        | 7.17         |
> > | 6    | WL-VH        | 7.33         |
> > | 7    | GraphSAGE    | 8.83         |
> > | 7    | GRF++ (ours)       | 8.83         |
> > | 7    | DiffPool     | 8.83         |
> > | 10   | DGCNN        | 9.50         |
> > | 11   | PM           | 10.33        |
> > | 12   | GRF          | 11.00        |
> > | 13   | ML           | 11.83        |
> > | 14   | GR           | 15.83        |
> > | 15   | ODD-STh      | 16.33        |
> > | 16   | CORE-SP      | 16.50        |
> > | 17   | SVM-θ        | 19.17        |
> > | 17   | PK           | 19.17        |
> > | 19   | SP           | 19.33        |
> > | 20   | WL-PM        | 19.50        |
> > | 21   | VH           | 21.50        |
> > | 22   | RW           | 22.17        |
> > | 23   | GH           | 23.17        |
> > | 24   | NSPDK        | 23.50        |
> > | 25   | WL-SP        | 24.17        |
> > | 26   | Lo-θ         | 25.33        |
> > | 27   | SM           | 27.00        |
> >
> >
> > [1] An End-to-End Deep Learning Architecture for Graph Classification. Zhang et al. AAAI 2018
> >
> > [2] Inductive representation learning on large graphs. Hamilton et al. NeurIPS 2017
> >
> > [3] Hierarchical graph representation learning with differentiable pooling. Ying et al. NeurIPS 2018
> >
> > [4] How Powerful are Graph Neural Networks? Xu et al. ICLR 2019
> >
> > [5] Graph Kernels: A Survey. Nikolentzos et al. JAIR 2021
> >
> >
> > **Speed Comparison**
> >
> > We highlight the speedups of GRF++ over the baseline brute force kernel method in the table below. A detailed plot showing our results over **40** meshes of various sizes are presented in Fig 7 in the updated version of the paper.
> >
> > | Mesh Size | BF Time (s) | GRF++ Time (s) | Speedup (BF / GRF++) |
> > |-----------:|------------:|---------------:|----------------------:|
> > | 2599       | 17.73       | 10.58          | **1.68× faster** |
> > | 2996       | 19.61       | 11.31          | **1.73× faster** |
> > | 3072       | 23.55       | 11.77          | **2.00× faster** |
> > | 3715       | 37.67       | 17.48          | **2.15× faster** |
> > | 5985       | 109.89      | 46.68          | **2.35× faster** |
> > | 6577       | 146.51      | 62.77          | **2.33× faster** |
> > | 7386       | 205.82      | 68.25          | **3.01× faster** |
> > | 7953       | 252.66      | 82.19          | **3.07× faster** |
> > | 8261       | 267.71      | 88.32          | **3.03× faster** |
> > | 9603       | 417.65      | 128.92         | **3.24× faster** |
> >
> > Conclusion: At moderate mesh sizes (2K–4K), GRF++ gives ~1.7×–2.2× speedup while for larger meshes (6K–10K), GRF++ consistently delivers ~3× faster runtime. Thus GRF++ scales significantly better as mesh complexity increases.

---

> > > ### Author Response · Authors · 2025-11-21
> > > **Answers to Reviewer's questions - part III**
> > >
> > > **Relationship to Nystrom Method:**
> > >
> > > Thank you for your question. Both GRFs and GRFs++ can in principle accommodate Nystrom method if regular signature vectors are replaced with their variations obtained by depositing loads only in the subset of selected anchor points. The anchor point technique was actually discussed before in the previous papers on GRFs.

---

> > > > ### Comment · Reviewer_r6Zd · 2025-11-26
> > > >
> > > > I would like to thank the authors for the response and for conducting the new experiments.
> > > >
> > > > However, I am still not convinced of the significance of the proposed GRF++ method.
> > > >
> > > > - In the node clustering results that the authors have provided, GRF++ outperforms all the baselines only on a single dataset. On Karate and Dolphins, KCenters achieves exactly the same performance as GRF++. On the Eurosis dataset, Spectral outperforms GRF++ (which is incorrectly highlighted in bold).
> > > >
> > > > - In the node classification experiments, GRF++ is outperformed by GCN on Cora Citeseer and Pubmed (according to the results presented in the original paper [1]). Notably, GCN is among the simplest GNN models.
> > > >
> > > > - GRF++ performs well in the graph classification experiments, but it is by no means a state-of-the-art method.
> > > >
> > > > Thank you for providing the speed comparison. GRF++ offers speed-ups over the brute-force algorithm. However, these speed-ups (~2–3×) are not that impressive, given that for most of the considered tasks (node clustering, classification, graph classification, mesh interpolation), computing the exact kernel can already be done within a few minutes.
> > > >
> > > > With regards to my previous comment on the Nystrom method, I was mainly asking whether this method could be used to speed-up the node kernels and serve as a baseline against GRF++. For instance, whether the diffusion kernel could be approximated using this approach instead of GRF++. To my understanding, this is not possible.
> > > >
> > > > [1] Kipf, T.N. & Welling, M. "Semi-supervised classification with graph convolutional networks." In 5th International Conference on Learning Representations, 2017.

---

### Official Review · Reviewer_UD4V · 2025-11-01

**Soundness:** 2
**Presentation:** 3
**Contribution:** 3
**Rating:** 2
**Confidence:** 4

**Summary:**

This work proposes GRFs++, a refinement of GRFs for unbiased approximation of node-level graph kernels. Two main ideas are presented; Walk Stitching of degree l; and general termination distributions. Theoretical results include an unbiasedness lemma for the stitched estimator with general terminations; a compact MSE identity for l=2; and a monotone MSE improvement under standard termination. Empirically, for the diffusion kernel only, GRFs++ reduce Frobenius error vs GRFs across synthetic and real graphs, improves distant-pair estimates, benefits further from Poisson termination and shoes useful downstream performance on graph classificaiton, node clustering and 3D mesh normal prediction.

**Strengths:**

(+) The observation that walk stitching corresponds to a 2l-fold de-convolution of the kernel's coefficient series is elegant and unifies construction.
(+) The termination-distribution generalization is a clean application of Russian-roulette style debiasing.
(+) From a parallelism & systems angle, short walks are easier to batch and parallelize on accelerators; stitching then composes them by MM products addressing a known pain point for GRFs where long RWs are inherently sequential.

**Weaknesses:**

(-)  The method is presented as applicable to a broad family of kernels, yet all experiments are diffusion-kernel only. To support generality please add a few more kernel classes and show the de-convolution construction of f and corresponding results. Also prior GRF variants are cited but not used as baselines. Adding comparisons against them would better situate GRF++ among the state of the art.
(-) Correctness issues: (i) Diffusion-kernel modulation f seems to be missing the lambda factor. In Sec 2.2.1 (in the paragraph starting at line 203) the paper states for diffusion kernels that the "correct" modulation is f(p) = 1/((2l)^p p!). Let G(x) = \sum_k a_k x^k = exp(\lambda x). For 2l-fold convolution we require F(x)^{2l} = G(x) ==>  F(x) = exp(\lambda/(2l) x). Therefore f(p) = ((\lambda/(2l))^p ) / p! . The missing \lambda^p seems to me to be a correctness bug that propagates to all diffusion-kernel experiments, unless \lambda was silently set to 1. Please fix the statement and clarify the value of \lambda used in the experiments. (ii) the termination pseudocode likely contains a sign error (since s_m is a sampled maximum length the 4th bullet point should be terminated <-- I[walk_length >= s_m], instead of "<="). (iii) In Lemma 3.2 in the derivation (step 5) uses "symmetry of X_1 and X_2" because "the (i, j) entry of each matrix is a dot-product of the random feature vectors ϕf (i) and ϕf (j), corresponding to vertices i and j". But with the factorization X_i = K_1^(i)(K_2)^(i) and independent K_1^(i), K_2^(i), X_i would not be generally symmetric. So... either define X_i as a gram matrix, or avoid the symmetry step. As written there seems to be a mismatch between the factorized form and the argument. Please clarify the exact objects and revise the proof accordingly.

Minor issues/typos:
- Duplicate reference to Beutel et al.
- walk-stitchng --> walk-stitching (page 6, line 288)
- standardize n vs N for sizes (page 5)

**Questions:**

Please provide answers to the correctness issues I list above

---

> ### Author Response · Authors · 2025-11-21
> **Answers to Reviewer's questions**
>
> We would like to sincerely thank the Reviewer for the feedback. We address all the questions below.
>
> **Going beyond diffusion kernels:**
>
> Thank you very much for the comment. We added results regarding GRF++ based approximation for three other graph kernels, in addition to the previously considered diffusion kernel (Section B.6). We also provided information regarding corresponding modulation functions. These results confirm all our previous findings for the diffusion kernel. **GRFs++ provide consistent accuracy gains**. **These results provide empirical validation of our theoretical analysis, showing that GRFs++ are not tailored to a particular graph kernel**.
>
> **Prior GRF variants:**
>
> The prior GRF variants, that the Reviewer is referring to, apply methods orthogonal to techniques presented in this paper (e.g. QMC approaches). Those techniques can be potentially combined with our methods rather than compared against.
>
> **Missing lambda-factor in the modulation factor:**
>
> Thank you very much for the comment. This is a typo and the  $\lambda$ factor is indeed needed. This typo is fixed now. It did not propagate to the experiments, where lambda is taken into account. It is done via the renormalized weighted adjacency matrix (that absorbs $\lambda$).
>
> **Termination strategy pseudocode:**
>
> Thank you very much for the comment. There was indeed a sign typo there that is fixed now.
>
> **Lemma 3.2:**
>
> Thank you very much for the comment. We have updated Lemma 3.2 so that no reliance on the symmetry of $X_{1}$, $X_{2}$ is needed anymore (see: the updated version of the paper).
>
> **Minor issues/typos:**
>
> These are all fixed now.

---

### Author Response · Authors · 2025-11-21
**General Comment**

We would like to sincerely thank the reviewers for the feedback and valuable comments. We address them in depth below. We have also made the following changes in the manuscript:

1. We have added speed tests comparing GRF++ with brute force kernel computations for graphs of varying sizes (Fig.7). They confirm our earlier findings. GRFs provide up to **3x** speedup.
2. We have added results regarding GRF++ based approximation for three other graph kernels, in addition to the previously considered diffusion kernel (Section B.6). We also provided information regarding corresponding modulation functions. These results confirm all our previous findings for the diffusion kernel. **GRFs++ provide consistent accuracy gains**. **These results provide empirical validation of our theoretical analysis, showing that GRFs++ are not tailored to a particular graph kernel**.
3. We have added 4 more baselines to our clustering results (Table 9).
4. **We have added 24 baselines to our graph classification results (Table 5, 6, 7)**. Moreover we have added graph classification experiments for two other datasets : Reddit-Multi-12k and Collab. GRFs++ provide strong performance, as compared to that updated comprehensive list of other methods.
5. We have integrated our GRF++ into Vision Transformer (ViT) models in Section B.5 (by re-interpreting images as having a 2-dimensional grid-graph structure) and **showed improvements over baseline ViT on ImageNet**.

---

### Author Response · Authors · 2025-12-02
**Note to ACs**

Dear ACs,

We appreciate the Area Chairs’ time and consideration, and we are grateful for the chance to address the reviewers’ concerns. To assist you in the final decision-making process, we would like to provide a concise summary of our core contributions and the significant updates made during the rebuttal period.

- We propose refined GRFs (GRFs++), which is a new class of Graph Random Features (GRFs) for efficient and accurate computations involving kernels defined on the nodes of a graph. We devise a novel walk stitching technique allowing us to unbiasedly estimate long walks by concatenating several shorter walks.
- We extend the GRFs walk termination mechanism (Bernoulli schemes with fixed halting probabilities) to a broader class of strategies, applying general distributions on the walks' lengths.
- We show that our method improves GRFs on a wide range of tasks like graph classification, node clustering and normal prediction.

During the rebuttal period at the request of the Reviewers, we added several experiments which are summarized below :

- We have integrated our GRF++ into Vision Transformer (ViT) models in Section B.5 (by re-interpreting images as having a 2-dimensional grid-graph structure) and showed improvements over baseline ViT on ImageNet.
- We have added results regarding GRF++ based approximation for three other graph kernels, in addition to the previously considered diffusion kernel (Section B.6). We also provided information regarding corresponding modulation functions. For all the kernels, GRFs++ provide consistent accuracy gains. These results provide empirical validation of our theoretical analysis, showing that GRFs++ are not tailored to a particular graph kernel.
- We have added speed tests comparing GRF++ with brute force kernel computations for graphs of varying sizes (Fig.7). GRFs provide up to **3x** speedup for larger graphs.
- We have added 4 more baselines to our clustering results (Table 9). GRF++ is competitive across all the baselines and the datasets.
- We have added **24** baselines to our graph classification results (Table 5, 6, 7). Moreover we have also added graph classification experiments for two other datasets : Reddit-Multi-12k and Collab. GRFs++ provide strong performance, as compared to that updated comprehensive list of other methods.

We would like to address the comparison with SOTA GNNs. We emphasize that Graph Kernels and GNNs occupy different computational niches. Kernels allow for distinct tradeoffs, such as convex optimization and rapid similarity search (e.g., via SVMs), which distinctively differ from end-to-end GNN training.

Furthermore, as our method is an approximation of a specific kernel, the exact kernel’s performance naturally acts as a theoretical upper bound. Our results show we approximate these kernels efficiently and accurately.

We sincerely appreciate the Area Chair’s careful attention and hope that our responses help clarify the motivation and relevance of our contribution.

---

### Meta-Review · Area_Chair_66KT · 2026-01-12

**Summary:**

1. My main concern with this paper is the significance of its contributions since the kernels between nodes that are approximated by the proposed GRFs++ are now rarely used in practice, having been largely replaced by GNNs that learn task-dependent features.

2. The paper provides a limited novelty. It extends the previously introduced g-GRF framework (Reid et al., 2024b) with a relatively incremental modification.

3. Almost all reviewers complained about lack of experimentation.

**Reviewer Concerns:**

The reviewers who had the time to reply weren't convinced with the author's replies.

**Reviewer Scores:**

The reviewers who had the time to reply weren't convinced with the author's replies.

---

### Decision · Program_Chairs · 2026-01-26

Reject